# Implementation of Learning Management Systems (LMS) in higher education systems through bipolar complex hesitant fuzzy Aczel-Alsina power aggregation operators: A case review for China

**Lijun Ma[1], Zeeshan Ali[2], Shi Yin[iD][3]***

**1** College of Land and Resources, Hebei Agricultural University, Baoding, China, **2** Department of Mathematics and Statistics, Riphah International University, Islamabad, Pakistan, **3** College of Economics and Management, Hebei Agriculture University, Baoding, China

* shyshi0314@163.com

**Data Availability Statement:** The authors confirm that the data supporting the findings of this study are available within the article.

## Abstract

A learning management system (LMS) is a web-based application or software platform computed to facilitate the development, tracking, management, reporting, and delivery of education and training programs. Many valuable and dominant factors are working behind the Learning Management System (LMS), but no one can find which factor is most important and valuable for LMS during COVID-19 among the following five alternatives, called Improved Accessibility, Blended Learning, Collaboration and Communications, Assessment and Evaluation, and Administrative Efficiency. For this, first, we derive the techniques of bipolar complex hesitant fuzzy (BCHF) sets, and then we evaluate some flexible operational laws, called Algebraic operational laws and Aczel-Alsina operational laws. Secondly, using the above techniques, we elaborate the technique of BCHF Aczel-Alsina power averaging (BCHFAAPA), BCHF Aczel-Alsina power weighted averaging (BCHFAAPWA), BCHF Aczel-Alsina power geometric (BCHFAAPG), and BCHF Aczel-Alsina power weighted geometric (BCHFAAPWG) operators. Some basic properties are also investigated for each proposed operator. Further, to evaluate the problem concerning LMS, we compute the multi-attribute decision-making (MADM) techniques for invented operators. Finally, we select some prevailing operators and try to compare their ranking results with our proposed results to enhance the worth and capability of the invented theory.

## 1. Introduction

LMS stands for Learning Management Systems. It's a software application or web-based technology considered to implement, assess a specific learning procedure, and plan to develop new techniques. The technique of LMS is mostly used for meeting or taking classes in government or private sector schools, colleges, and universities to facilitate learning in many formats, such

**Funding:** The author(s) received no specific funding for this work.

**Competing interests:** The authors declare no conflict of interest.

as online courses, virtual classrooms, assessments, communications, and reporting tools. The implementation of the LMS in every government and private sector is very important because, during COVID-19, many sectors have faced a lot of problems due to some complications and problems. With the help of LMS techniques, everyone can easily deliver his class and also do his meetings with others. Many applications have been done by different scholars based on the utilization of the LMSs, but because of classical information, every expert has lost a lot of information because of limited opinions. To increase the range of opinions, Zadeh [1] evaluated the novel technique of fuzzy set (FS), where the FS has only a positive grade whose values are limited to the unit interval, but many experts talked about the negative membership because negative membership grade is also a very important part of many problems, therefore, the bipolar fuzzy set (BFS) was presented by Zhang [2]. BFS talked about the positive and negative membership grades, where the range of both grades is different or in opposite directions. Many applications of BFS have been initiated by a lot of individuals, for instance, type-2 generalized Fermatean BFSs [3], bipolar fuzzy metric spaces [4], 2-tuple linguistic BFSs [5], bipolar vague soft sets [6], analysis of different types of classes for bipolar soft sets [7], TOPSIS techniques for bipolar soft covering sets [8], ideals of semiring based on bipolar techniques [9], and interval BFSs and their application in the environment of TOPSIS methods [10].

A positive grade is not enough for handling ambiguous and unreliable information, because in many cases experts have faced complicated problems that cannot handled from one-dimension positive grades. For this, we have needed strong and valid techniques whose positive grades are computed in the shape of complex numbers or two-dimensional information. Therefore, the idea of a complex fuzzy set (CFS) was initiated by Ramot et al. [11], where the grade in CFS is computed in the shape of complex numbers, whose real and unreal parts are defined from universal set to unit interval. Furthermore, Mahmood and Ur Rehman [12] initiated the technique of bipolar complex fuzzy set (BCFS), where the technique of BCFS is very reliable because it contains the positive and negative membership grades in the shape of complex numbers. BCFS is a very dominant technique due to its structure, where the FSs, CFSs, and BFSs are some special cases of the BCFSs. Furthermore, Liu et al. [13] exposed the cross-entropy measures based on CFSs, and Mahmood et al. [14] evaluated the interdependency of complex fuzzy operators. Recently, Mahmood and Ali [15] derived the complex fuzzy N-soft sets and their application in decision-making problems.

Torra [16] initiated the technique of hesitant fuzzy set (HFS), where the positive grade in HFS is computed in the form of a group or collection of fuzzy membership grades because in many situations we have required the membership grades in the shape of the group. Furthermore, the bipolar hesitant fuzzy set (BHFS) was initiated by Ullah et al. [17]. The FSs and HFSs are the special cases of the BHFS, but still, the positive grade is not enough to deal with ambiguous and unreliable data. For this, complex HFS (CHFS) was initiated by Mahmood et al. [18], where the positive grades in CHFS are arranged in the shape of complex numbers. Many applications of HFSs and CHFSs were computed by different scholars, for instance, exponential and non-exponential measures for CHFSs [19], cosine similarity measures for CHFSs [20], and Jaccard and dice similarity measures for CHFSs [21]. Furthermore, for aggregating the collection of data, we have required strong and dominant ideas, which can help in the construction of aggregation operators, for this, Aczel and Alsina [22] evaluated the novel idea of Aczel-Alsina t-norm and t-conorm based on classical set theory which is the modified version of the algebraic norms. Moreover, Wang et al. [23] derived the multi-biometric fusion based on Aczel-Alsina norms, and Pamucar et al. [24] exposed the Aczel-Alsina function based on fuzzy rough sets.

In the above paragraphs, we discussed all the possibilities of the utilization of the Aczel-Alsina power operators based on different ideas. After a long discussion, we noticed that no

one could derive the idea of power aggregation operators [25] based on Aczel-Alsina operational laws. Because many scholars have utilized the Aczel-Alsina operators in the environment of different fields, for instance, Senapati et al. [26] derived the Aczel-Alsina operators for HFSs, Mahmood et al. [27] evaluated the Aczel-Alsina operators for BCFSs, Garg et al. [28] invented the Aczel-Alsina power operators for BFSs, and Mahmood et al. [29] examined the geometric Aczel-Alsina operators for BCFSs. After our brief assessments, we observed that a lot of experts have faced the following three major problems during the decision-making procedure, such as

1. Evaluating new ideas is very complex.

2. To aggregate the collection of information into a singleton set is also very complex.

3. Finding the best optimal among the collection of finite information is very awkward.

But to address these queries is very simple if we propose the novel theory of BCHFSs, Aczel-Alsina operational laws, Aczel-Alsina power averaging/geometric operators, and their properties. After a long discussion, we also noticed that no one could evaluate the idea of BCHFSs because it is very complex. The BCHFS is the combination of many ideas, called FSs, HFSs, CFSs, and BCFSs, because of this reason, many scholars have failed to develop it. Keeping these problems in mind, we start the construction of the BCHFSs in this manuscript because of their advantages, where the advantages of the proposed theory are listed below:

1. The special case of the proposed theory: Averaging operator and geometric operator for FSs to BCHFSs.

2. The special case of the proposed theory: Power averaging operator and power geometric operator for FSs to BCHFSs.

3. The special case of the proposed theory: Aczel-Alsina averaging operator and Aczel-Alsina geometric operator for FSs to BCHFSs.

4. The special case of the proposed theory: Aczel-Alsina power averaging operator and Aczel-Alsina power geometric operator for FSs to BCHFSs.

For the above analysis, it is given that many operators based on different ideas are the special cases of the proposed theory, where the FSs, CFSs, HFSs, CHFSs, BFSs, BCFSs, Bipolar HFSs, bipolar complex HFSs are the special cases of the BCHFSs. Keeping the advantages of the proposed theory, the major contribution of the proposed article is listed below:

1. To derive the techniques of BCHF sets, then we evaluate some flexible operational laws, called Algebraic operational laws and Aczel-Alsina operational laws.

2. To elaborate on the techniques of BCHFAAPA, BCHFAAPWA, BCHFAAPG, and BCHFAAPWG operators.

3. To discuss some basic properties of each proposed operator.

4. To compute the MADM techniques for invented operators.

5. To select some prevailing operators and try to compare their ranking results with our proposed results to enhance the worth and capability of the invented theory.

Our proposed article is arranged in the shape: In Section 2, we stated the power aggregation operators (PAO), BHFSs, and their operational laws. In Section 3, we described the novel techniques of BCHF sets and their flexible operational laws, especially Aczel-Alsina operational laws. In Section 4, we examined the BCHFAAPA operator, BCHFAAPWA operator,

BCHFAAPG operator, and BCHFAAPWG operator, and also derived their basic properties. In Section 5, we evaluated the major source that plays an important role in the utilization of LMSs, we demonstrated the technique of MADM techniques based on the invented operators for BCHF information. In Section 6, we selected some prevailing operators and tried to compare their ranking results with our proposed results to enhance the depth of the derived theory. some concluding remarks are derived in Section 7.

## 2. Preliminaries

For a universal set $\mathbb{X}$, we discussed the idea of BHFSs and their fundamental laws with a crip Aczel-Alsina t-norm and t-conorm. Additionally, we have also started the idea of PAOs [25] based on classical information, such as, for any collection of non-negative integers, the basic idea of PAOs is derived by:

$$PA\left(\mathfrak{Y}_{BF}^1, \mathfrak{Y}_{BF}^2, \ldots, \mathfrak{Y}_{BF}^n\right) = \oplus_{k=1}^q \left(\frac{1 +^\circ \mathrm{C}\left(\mathfrak{Y}_{BF}^k\right)}{\sum_{k=1}^q \left(1 +^\circ \mathrm{C}\left(\mathfrak{Y}_{BF}^k\right)\right)}\right)\mathfrak{Y}_{BF}^k$$

With some information, such as $^\circ\mathrm{C}\left(\mathfrak{Y}_{BF}^k\right) = \sum_{l=1}^p SUP\left(\mathfrak{Y}_{BF}^k, \mathfrak{Y}_{BF}^l\right)$ and $SUP\left(\mathfrak{Y}_{BF}^k, \mathfrak{Y}_{BF}^l\right) = 1 - DIS\left(\mathfrak{Y}_{BF}^k, \mathfrak{Y}_{BF}^l\right)$, with some characteristics, such as

1. $SUP\left(\mathfrak{Y}_{BF}^k, \mathfrak{Y}_{BF}^l\right) \in [0, 1]$.

2. $SUP\left(\mathfrak{Y}_{BF}^k, \mathfrak{Y}_{BF}^l\right) = SUP\left(\mathfrak{Y}_{BF}^l, \mathfrak{Y}_{BF}^k\right)$.

3. If $SUP\left(\mathfrak{Y}_{BF}^k, \mathfrak{Y}_{BF}^l\right) \geq SUP\left(\mathfrak{Y}_{BF}^m, \mathfrak{Y}_{BF}^n\right)$ then $DIS\left(\mathfrak{Y}_{BF}^k, \mathfrak{Y}_{BF}^l\right) \leq DIS\left(\mathfrak{Y}_{BF}^m, \mathfrak{Y}_{BF}^n\right)$.

**Definition 1**: [17] Consider a universal set $\mathbb{X}$. The idea of BHFS $\mathfrak{Y}_{BF}$ is invented by:

$$\mathfrak{Y}_{BF} = \left\{\left(\mathbb{T}_{MS}(\mathbb{x}), \mathbb{F}_{NMS}(\mathbb{x})\right) : \mathbb{x} \in \mathbb{X}\right\}$$

Where the term $\mathbb{T}_{MS}(\mathbb{x}) = \left\{\mathbb{T}_{RP}^j(\mathbb{x}) : j = 1, 2, \ldots, z\right\}$ and $\mathbb{F}_{NMS}(\mathbb{x}) = \left\{\mathbb{F}_{RP}^j(\mathbb{x}) : j = 1, 2, \ldots, z\right\}$ shows the positive and negative truth grades with a strategy: $\mathbb{T}_{RP}^j : \mathbb{X} \to [0, 1]$ and $\mathbb{F}_{RP}^j : \mathbb{X} \to [-1, 0]$. Furthermore, the simple shape of BHF number (BHFN) is stated by: $\mathfrak{Y}_{BF}^k = \left(\mathbb{T}_{MS}, \mathbb{F}_{NMS}\right) = \left(\left\{\mathbb{T}_{RP}^j\right\}, \left\{\mathbb{F}_{RP}^j\right\}\right), k = 1, 2, \ldots, q$.

**Definition 2**: [17] For any two BHFNs $\mathfrak{Y}_{BF}^k = \left(\mathbb{T}_{MS}, \mathbb{F}_{NMS}\right) = \left(\mathbb{T}_{RP_k}^j, \mathbb{F}_{RP_k}^j\right), k = 1, 2$, we have

$$\mathfrak{Y}_{BF}^1 \oplus \mathfrak{Y}_{BF}^2 = \coprod_{\begin{pmatrix} \mathbb{T}_{RP}^1, \mathbb{T}_{RP}^2 \in \mathbb{T}_{MS}, \\ \mathbb{F}_{RP}^1, \mathbb{F}_{RP}^2 \in \mathbb{F}_{NMS} \end{pmatrix}} \begin{pmatrix} \left(\mathbb{T}_{RP}^j + \mathbb{T}_{RP}^j - \mathbb{T}_{RP}^j \mathbb{T}_{RP}^j\right), \\ -\left(\mathbb{F}_{RN}^j \mathbb{F}_{RN}^j\right) \end{pmatrix}$$

$$\mathfrak{Y}_{BF}^1 \otimes \mathfrak{Y}_{BF}^2 = \coprod_{\begin{pmatrix} \mathbb{T}_{RP}^1, \mathbb{T}_{RP}^2 \in \mathbb{T}_{MS}, \\ \mathbb{F}_{RP}^1, \mathbb{F}_{RP}^2 \in \mathbb{F}_{NMS} \end{pmatrix}} \begin{pmatrix} \left(\mathbb{T}_{RP}^1 \mathbb{T}_{RP}^2\right), \\ \left(\mathbb{F}_{RN}^1 + \mathbb{F}_{RN}^2 + \mathbb{F}_{RN}^1 \mathbb{F}_{RN}^2\right) \end{pmatrix}$$

$$\varpi_s^c \mathfrak{Y}_{BF}^1 = \coprod_{\left(\begin{array}{c} \mathbb{T}_{RP}^1, \mathbb{T}_{RP}^2 \in \mathbb{T}_{MS}, \\ \mathbb{F}_{RP}^1, \mathbb{F}_{RP}^2 \in \mathbb{F}_{NMS} \end{array}\right)} \left( 1 - \left( 1 - \mathbb{T}_{RP}^1 \right)^{\varpi_s^c}, -\left| \mathbb{F}_{RN}^1 \right|^{\varpi_s^c} \right)$$

$$\left( \mathfrak{Y}_{BF}^1 \right)^{\varpi_s^c} = \coprod_{\left(\begin{array}{c} \mathbb{T}_{RP}^1, \mathbb{T}_{RP}^2 \in \mathbb{T}_{MS}, \\ \mathbb{F}_{RP}^1, \mathbb{F}_{RP}^2 \in \mathbb{F}_{NMS} \end{array}\right)} \left( \left( \mathbb{T}_{RP}^1 \right)^{\varpi_s^c}, -1 + \left( 1 + \mathbb{F}_{RN}^1 \right)^{\varpi_s^c} \right)$$

**Definition 3**: [17] For any two BHFNs, we have

$$S\left( \mathfrak{Y}_{BF}^1 \right) = \frac{1}{order\ of\ \left( \mathbb{T}_{RP}^j \right)} \sum_{j=1}^z \mathbb{T}_{RP}^j - \frac{1}{order\ of\ \left( \mathbb{F}_{RP}^j \right)} \sum_{j=1}^z \mathbb{F}_{RP}^j \in [-1, 1]$$

$$H\left( \mathfrak{Y}_{BF}^1 \right) = \frac{1}{order\ of\ \left( \mathbb{T}_{RP}^j \right)} \sum_{j=1}^z \mathbb{T}_{RP}^j + \frac{1}{order\ of\ \left( \mathbb{F}_{RP}^j \right)} \sum_{j=1}^z \mathbb{F}_{RP}^j \in [0, 1]$$

Some fundamental characteristics are stated below: when $S\left( \mathfrak{Y}_{BF}^1 \right) > S\left( \mathfrak{Y}_{BF}^2 \right)$, then $\mathfrak{Y}_{BF}^1 > \mathfrak{Y}_{BF}^2$, when $S\left( \mathfrak{Y}_{BF}^1 \right) < S\left( \mathfrak{Y}_{BF}^2 \right)$, then $\mathfrak{Y}_{BF}^1 < \mathfrak{Y}_{BF}^2$, when $S\left( \mathfrak{Y}_{BF}^1 \right) = S\left( \mathfrak{Y}_{BF}^2 \right)$, then, when $H\left( \mathfrak{Y}_{BF}^1 \right) > H\left( \mathfrak{Y}_{BF}^2 \right)$, then $\mathfrak{Y}_{BF}^1 > \mathfrak{Y}_{BF}^2$, when $H\left( \mathfrak{Y}_{BF}^1 \right) < H\left( \mathfrak{Y}_{BF}^2 \right)$, then $\mathfrak{Y}_{BF}^1 < \mathfrak{Y}_{BF}^2$.

**Definition 4**: [22] An Aczel-Alsina t-norm is invented by:

$$\overline{\overline{\mathbb{T}_{TN}}}^\Xi (\mathbb{x}_1, \mathbb{x}_2) = \begin{cases} \overline{\overline{\mathbb{T}_{tn}}}(\mathbb{x}_1, \mathbb{x}_2) & when\ \Xi = 0 \\ min(\mathbb{x}_1, \mathbb{x}_2) & when\ \Xi = \infty \\ e^{-\left( (-\ln(\mathbb{x}_1))^\Xi + (-\ln(\mathbb{x}_2))^\Xi \right)^{\frac{1}{\Xi}}} & otherwise \end{cases}$$

An Aczel-Alsina t-conorm is derived by:

$$\overline{\overline{\mathbb{S}_{TCN}}}^\Xi (\mathbb{x}_1, \mathbb{x}_2) = \begin{cases} \overline{\overline{\mathbb{S}_{tcn}}}(\mathbb{x}_1, \mathbb{x}_2) & if\ \Xi = 0 \\ max(\mathbb{x}_1, \mathbb{x}_2) & if\ \Xi = \infty \\ 1 - e^{-\left( (-\ln(1-\mathbb{x}_1))^\Xi + (-\ln(1-\mathbb{x}_2))^\Xi \right)^{\frac{1}{\Xi}}} & otherwise \end{cases}$$

Where $\overline{\overline{\mathbb{T}_{tn}}} : [0, 1] \times [0, 1] \to [0, 1]$ is a function with $\Xi \geq 0$. Furthermore, we derive the following information from the above theory, such as $\overline{\overline{\mathbb{T}_{tn}}}(\mathbb{x}_1, \mathbb{x}_2) = \mathbb{x}_1 * \mathbb{x}_2$ and $\overline{\overline{\mathbb{S}_{tcn}}}(\mathbb{x}_1, \mathbb{x}_2) = \mathbb{x}_1 + \mathbb{x}_2 - \mathbb{x}_1 * \mathbb{x}_2$.

## 3. Bipolar complex hesitant fuzzy sets

In this section, we invent the theory of BCHFSs and their fundamental laws such as Algebraic laws and Aczel-Alsina laws.

**Definition 5:** Consider a universal set $\mathbb{X}$. The idea of BCHFS $\mathfrak{Y}_{BF}$ is invented by:

$$\mathfrak{Y}_{BF} = \left\{ \left( \mathbb{T}_{MS}(\mathbb{x}), \mathbb{F}_{NMS}(\mathbb{x}) \right) : \mathbb{x} \in \mathbb{X} \right\}$$

Where the term $\mathbb{T}_{MS}(\mathbb{x}) = \left\{ \mathbb{T}_{RP}^j(\mathbb{x}) + i\mathbb{T}_{IP}^j(\mathbb{x}) : j = 1, 2, \ldots, z \right\}$ and $\mathbb{F}_{NMS}(\mathbb{x}) = \left\{ \mathbb{F}_{RP}^j(\mathbb{x}) + i\mathbb{F}_{IP}^j(\mathbb{x}) : j = 1, 2, \ldots, z \right\}$ shows the positive and negative truth grades with a strategy: $\mathbb{T}_{RP}^j, \mathbb{T}_{IP}^j : \mathbb{X} \to [0, 1]$ and $\mathbb{F}_{RP}^j, \mathbb{F}_{IP}^j : \mathbb{X} \to [-1, 0]$. Furthermore, the simple shape of the

BCHF number (BCHFN) is stated by:

$$\mathfrak{Y}^k_{BF} = (\mathbb{T}_{MS}, \mathbb{F}_{NMS}) = \left(\left\{\mathbb{T}^j_{RP} + \mathring{\mathbb{i}}\mathbb{T}^j_{RP}\right\}, \left\{\mathbb{F}^j_{RP} + \mathring{\mathbb{i}}\mathbb{F}^j_{RP}\right\}\right), k = 1, 2, \ldots, q.$$

**Definition 6:** For any two BCHFNs

$$\mathfrak{Y}^k_{BF} = (\mathbb{T}_{MS}, \mathbb{F}_{NMS}) = \left(\left\{\mathbb{T}^j_{RP} + \mathring{\mathbb{i}}\mathbb{T}^j_{RP}\right\}, \left\{\mathbb{F}^j_{RP} + \mathring{\mathbb{i}}\mathbb{F}^j_{RP}\right\}\right), k = 1, 2, \text{ we have}$$

$$\mathfrak{Y}^1_{BF} \oplus \mathfrak{Y}^2_{BF} = \coprod_{\begin{pmatrix} \mathbb{T}^1_{RP}, \mathbb{T}^2_{RP}, \mathbb{T}^1_{IP}, \mathbb{T}^2_{IP} \in \mathbb{T}_{MS}, \\ \mathbb{F}^1_{RP}, \mathbb{F}^2_{RP}, \mathbb{F}^1_{IP}, \mathbb{F}^2_{IP} \in \mathbb{F}_{NMS} \end{pmatrix}} \begin{pmatrix} \left(\mathbb{T}^1_{RP} + \mathbb{T}^2_{RP} - \mathbb{T}^1_{RP}\mathbb{T}^2_{RP}\right) + \mathring{\mathbb{i}}\left(\mathbb{T}^1_{IP} + \mathbb{T}^2_{IP} - \mathbb{T}^1_{IP}\mathbb{T}^2_{IP}\right), \\ -\left(\mathbb{F}^1_{RN}\mathbb{F}^2_{RN}\right) + \mathring{\mathbb{i}}\left(-\left(\mathbb{F}^1_{IN}\mathbb{F}^2_{IN}\right)\right) \end{pmatrix}$$

$$\mathfrak{Y}^1_{BF} \otimes \mathfrak{Y}^2_{BF} = \coprod_{\begin{pmatrix} \mathbb{T}^1_{RP}, \mathbb{T}^2_{RP}, \mathbb{T}^1_{IP}, \mathbb{T}^2_{IP} \in \mathbb{T}_{MS}, \\ \mathbb{F}^1_{RP}, \mathbb{F}^2_{RP}, \mathbb{F}^1_{IP}, \mathbb{F}^2_{IP} \in \mathbb{F}_{NMS} \end{pmatrix}} \begin{pmatrix} \left(\mathbb{T}^1_{RP}\mathbb{T}^2_{RP}\right) + \mathring{\mathbb{i}}\left(\mathbb{T}^1_{IP}\mathbb{T}^2_{IP}\right), \\ \left(\mathbb{F}^1_{RN} + \mathbb{F}^2_{RN} + \mathbb{F}^1_{RN}\mathbb{F}^2_{RN}\right) + \mathring{\mathbb{i}}\left(\mathbb{F}^1_{IN} + \mathbb{F}^2_{IN} + \mathbb{F}^1_{IN}\mathbb{F}^2_{IN}\right) \end{pmatrix}$$

$$\varpi^c_s \mathfrak{Y}^1_{BF} = \coprod_{\begin{pmatrix} \mathbb{T}^1_{RP}, \mathbb{T}^1_{IP} \in \mathbb{T}_{MS}, \\ \mathbb{F}^1_{RP}, \mathbb{F}^1_{IP} \in \mathbb{F}_{NMS} \end{pmatrix}} \left(1 - \left(1 - \mathbb{T}^1_{RP}\right)^{\varpi^c_s} + \mathring{\mathbb{i}}\left(1 - \left(1 - \mathbb{T}^1_{IP}\right)^{\varpi^c_s}\right), -|\mathbb{F}^1_{RN}|^{\varpi^c_s} + \mathring{\mathbb{i}}\left(-\mathbb{F}^1_{IN}\varpi^c_s\right)\right)$$

$$\left(\mathfrak{Y}^1_{BF}\right)^{\varpi^c_s} = \coprod_{\begin{pmatrix} \mathbb{T}^1_{RP}, \mathbb{T}^1_{IP} \in \mathbb{T}_{MS}, \\ \mathbb{F}^1_{RP}, \mathbb{F}^1_{IP} \in \mathbb{F}_{NMS} \end{pmatrix}} \left(\left(\mathbb{T}^1_{RP}\right)^{\varpi^c_s} + \mathring{\mathbb{i}}\left(\mathbb{T}^1_{IP}\right)^{\varpi^c_s}, -1 + \left(1 + \mathbb{F}^1_{RN}\right)^{\varpi^c_s} + \mathring{\mathbb{i}}\left(-1 + \left(1 + \mathbb{F}^1_{IN}\right)^{\varpi^c_s}\right)\right)$$

**Definition 7:** For any two BCHFNs, we have

$$S\left(\mathfrak{Y}^1_{BF}\right) = \frac{1}{2}\left(\begin{array}{c} \dfrac{1}{order\ of\ \left(\mathbb{T}^j_{RP}\right)}\displaystyle\sum_{j=1}^z \mathbb{T}^j_{RP} + \dfrac{1}{order\ of\ \left(\mathbb{T}^j_{IP}\right)}\displaystyle\sum_{j=1}^z \mathbb{T}^j_{IP} - \\ \dfrac{1}{order\ of\ \left(\mathbb{F}^j_{RP}\right)}\displaystyle\sum_{j=1}^z \mathbb{F}^j_{RP} - \dfrac{1}{order\ of\ \left(\mathbb{F}^j_{IP}\right)}\displaystyle\sum_{j=1}^z \mathbb{F}^j_{IP} \end{array}\right) \in [-1, 1]$$

$$H\left(\mathfrak{Y}^1_{BF}\right) = \frac{1}{2}\left(\begin{array}{c} \dfrac{1}{order\ of\ \left(\mathbb{T}^j_{RP}\right)}\displaystyle\sum_{j=1}^z \mathbb{T}^j_{RP} + \dfrac{1}{order\ of\ \left(\mathbb{T}^j_{IP}\right)}\displaystyle\sum_{j=1}^z \mathbb{T}^j_{IP} + \\ \dfrac{1}{order\ of\ \left(\mathbb{F}^j_{RP}\right)}\displaystyle\sum_{j=1}^z \mathbb{F}^j_{RP} + \dfrac{1}{order\ of\ \left(\mathbb{F}^j_{IP}\right)}\displaystyle\sum_{j=1}^z \mathbb{F}^j_{IP} \end{array}\right) \in [0, 1]$$

Some fundamental characteristics are stated below: when $S\left(\mathfrak{Y}^1_{BF}\right) > S\left(\mathfrak{Y}^2_{BF}\right)$, then $\mathfrak{Y}^1_{BF} > \mathfrak{Y}^2_{BF}$, when $S\left(\mathfrak{Y}^1_{BF}\right) < S\left(\mathfrak{Y}^2_{BF}\right)$, then $\mathfrak{Y}^1_{BF} < \mathfrak{Y}^2_{BF}$, when $S\left(\mathfrak{Y}^1_{BF}\right) = S\left(\mathfrak{Y}^2_{BF}\right)$, then, when $H\left(\mathfrak{Y}^1_{BF}\right) > H\left(\mathfrak{Y}^2_{BF}\right)$, then $\mathfrak{Y}^1_{BF} > \mathfrak{Y}^2_{BF}$, when $H\left(\mathfrak{Y}^1_{BF}\right) < H\left(\mathfrak{Y}^2_{BF}\right)$, then $\mathfrak{Y}^1_{BF} < \mathfrak{Y}^2_{BF}$.

**Definition 8:** For any two BCHFNs

$$\mathfrak{Y}_{BF}^k = (\mathbb{T}_{MS}, \mathbb{F}_{NMS}) = \left( \left\{ \mathbb{T}_{RP}^j + i\mathbb{T}_{RP}^j \right\}, \left\{ \mathbb{F}_{RP}^j + i\mathbb{F}_{RP}^j \right\} \right), k = 1, 2, \text{ we have}$$

$$\mathfrak{Y}_{BF}^1 \oplus \mathfrak{Y}_{BF}^2 = \coprod_{\begin{pmatrix} \mathbb{T}_{RP}^1, \mathbb{T}_{RP}^2, \mathbb{T}_{IP}^1, \mathbb{T}_{IP}^2 \in \mathbb{T}_{MS}, \\ \mathbb{F}_{RP}^1, \mathbb{F}_{RP}^2, \mathbb{F}_{IP}^1, \mathbb{F}_{IP}^2 \in \mathbb{F}_{NMS} \end{pmatrix}}$$

$$\begin{pmatrix} 1 - e^{-\left( \left( -\ln\left(1-\mathbb{T}_{RP}^1\right)\right)^{\Xi} + \left( -\ln\left(1-\mathbb{T}_{RP}^2\right)\right)^{\Xi} \right)^{\frac{1}{\Xi}}} + i\left( 1 - e^{-\left( \left( -\ln\left(1-\mathbb{T}_{IP}^1\right)\right)^{\Xi} + \left( -\ln\left(1-\mathbb{T}_{IP}^2\right)\right)^{\Xi} \right)^{\frac{1}{\Xi}}} \right), \\ -\left( e^{-\left( \left( -\ln\left(\mathbb{F}_{RN}^1\right)\right)^{\Xi} + \left( -\ln\left(\mathbb{F}_{RN}^2\right)\right)^{\Xi} \right)^{\frac{1}{\Xi}}} \right) + i\left( -\left( e^{-\left( \left( -\ln\left(\left|\mathbb{F}_{IN}^1\right|\right)\right)^{\Xi} + \left( -\ln\left(\left|\mathbb{F}_{IN}^2\right|\right)\right)^{\Xi} \right)^{\frac{1}{\Xi}}} \right) \right) \end{pmatrix}$$

$$\mathfrak{Y}_{BF}^1 \otimes \mathfrak{Y}_{BF}^2 = \coprod_{\begin{pmatrix} \mathbb{T}_{RP}^1, \mathbb{T}_{RP}^2, \mathbb{T}_{IP}^1, \mathbb{T}_{IP}^2 \in \mathbb{T}_{MS}, \\ \mathbb{F}_{RP}^1, \mathbb{F}_{RP}^2, \mathbb{F}_{IP}^1, \mathbb{F}_{IP}^2 \in \mathbb{F}_{NMS} \end{pmatrix}}$$

$$\begin{pmatrix} e^{-\left( \left( -\ln\left(\mathbb{T}_{RP}^1\right)\right)^{\Xi} + \left( -\ln\left(\mathbb{T}_{RP}^2\right)\right)^{\Xi} \right)^{\frac{1}{\Xi}}} + i\left( e^{-\left( \left( -\ln\left(\mathbb{T}_{IP}^1\right)\right)^{\Xi} + \left( -\ln\left(\mathbb{T}_{IP}^2\right)\right)^{\Xi} \right)^{\frac{1}{\Xi}}} \right), \\ -1 + \left( e^{-\left( \left( -\ln\left(\left(1+\mathbb{F}_{RN}^1\right)\right)\right)^{\Xi} + \left( -\ln\left(\left(1+\mathbb{F}_{RN}^2\right)\right)\right)^{\Xi} \right)^{\frac{1}{\Xi}}} \right) + i\left( -1 + \left( e^{-\left( \left( -\ln\left(\left(1+\mathbb{F}_{IN}^1\right)\right)\right)^{\Xi} + \left( -\ln\left(\left(1+\mathbb{F}_{IN}^2\right)\right)\right)^{\Xi} \right)^{\frac{1}{\Xi}}} \right) \right) \end{pmatrix}$$

$$\varpi_s^c \mathfrak{Y}_{BF}^1 = \coprod_{\begin{pmatrix} \mathbb{T}_{RP}^1, \mathbb{T}_{IP}^1 \in \mathbb{T}_{MS}, \\ \mathbb{F}_{RP}^1, \mathbb{F}_{IP}^1 \in \mathbb{F}_{NMS} \end{pmatrix}}$$

$$\begin{pmatrix} 1 - e^{-\left( \varpi_s^c \left( -\ln\left(1-\mathbb{T}_{RP}^1\right)\right)^{\Xi} \right)^{\frac{1}{\Xi}}} + i\left( 1 - e^{-\left( \varpi_s^c \left( -\ln\left(1-\mathbb{T}_{IP}^1\right)\right)^{\Xi} \right)^{\frac{1}{\Xi}}} \right), \\ -\left( e^{-\left( \varpi_s^c \left( -\ln\left(\mathbb{F}_{RN}^1\right)\right)^{\Xi} \right)^{\frac{1}{\Xi}}} \right) + i\left( -\left( e^{-\left( \varpi_s^c \left( -\ln\left(\left|\mathbb{F}_{IN}^1\right|\right)\right)^{\Xi} \right)^{\frac{1}{\Xi}}} \right) \right) \end{pmatrix}, \varpi_s^c \geq 1$$

$$\left(\mathfrak{Y}_{BF}^1\right)^{\varpi_s^c} = \coprod_{\begin{pmatrix} \mathbb{T}_{RP}^1, \mathbb{T}_{IP}^1 \in \mathbb{T}_{MS}, \\ \mathbb{F}_{RP}^1, \mathbb{F}_{IP}^1 \in \mathbb{F}_{NMS} \end{pmatrix}}$$

$$\begin{pmatrix} e^{-\left( \varpi_s^c \left( -\ln\left(\mathbb{T}_{RP}^1\right)\right)^{\Xi} \right)^{\frac{1}{\Xi}}} + i\left( e^{-\left( \varpi_s^c \left( -\ln\left(\mathbb{T}_{IP}^1\right)\right)^{\Xi} \right)^{\frac{1}{\Xi}}} \right), \\ -1 + \left( e^{-\left( \varpi_s^c \left( -\ln\left(\left(1+\mathbb{F}_{RN}^1\right)\right)\right)^{\Xi} \right)^{\frac{1}{\Xi}}} \right) + i\left( -1 + \left( e^{-\left( \varpi_s^c \left( -\ln\left(\left(1+\mathbb{F}_{IN}^1\right)\right)\right)^{\Xi} \right)^{\frac{1}{\Xi}}} \right) \right) \end{pmatrix}$$

## 4. Aczel-Alsina power aggregation operators for BCHFSs

In this section, we expose the novel theory of the BCHFAAPA operator, BCHFAAPWA operator, BCHFAAPG operator, and BCHFAAPWG operator, and describe their valuable

properties such as idempotency, monotonicity, and boundedness. In this section, we used the BCHFNs by: $\mathfrak{Y}_{BF}^k = (\mathbb{T}_{MS}, \mathbb{F}_{NMS}) = \left(\left\{\mathbb{T}_{RP}^j + \mathring{\mathbb{i}}\mathbb{T}_{RP}^j\right\}, \left\{\mathbb{F}_{RP}^j + \mathring{\mathbb{i}}\mathbb{F}_{RP}^j\right\}\right), k = 1, 2, \ldots, q.$

**Definition 9:** For any collection of BCHFNs, the BCHFAAPA operator is presented and arranged in the shape:

$$BCHFAAPA\left(\mathfrak{Y}_{BF}^1, \mathfrak{Y}_{BF}^2, \ldots, \mathfrak{Y}_{BF}^q\right) = \left(\frac{1 +^\circ C\left(\mathfrak{Y}_{BF}^1\right)}{\sum_{k=1}^q \left(1 +^\circ C\left(\mathfrak{Y}_{BF}^k\right)\right)}\right)\mathfrak{Y}_{BF}^1$$

$$\oplus \left(\frac{1 +^\circ C\left(\mathfrak{Y}_{BF}^2\right)}{\sum_{k=1}^q \left(1 +^\circ C\left(\mathfrak{Y}_{BF}^k\right)\right)}\right)\mathfrak{Y}_{BF}^2 \oplus \ldots \oplus \left(\frac{1 +^\circ C\left(\mathfrak{Y}_{BF}^q\right)}{\sum_{k=1}^q \left(1 +^\circ C\left(\mathfrak{Y}_{BF}^k\right)\right)}\right)\mathfrak{Y}_{BF}^q$$

$$= \sum_{k=1}^q \left(\frac{1 +^\circ C\left(\mathfrak{Y}_{BF}^k\right)}{\sum_{k=1}^q \left(1 +^\circ C\left(\mathfrak{Y}_{BF}^k\right)\right)}\right)\mathfrak{Y}_{BF}^k$$

**Theorem 1:** For any collection of BCHFNs, then we prove that the aggregated value of the above information is again a BCHFN, such as

$$BCHFAAPA\left(\mathfrak{Y}_{BF}^1, \mathfrak{Y}_{BF}^2, \ldots, \mathfrak{Y}_{BF}^q\right) = \coprod_{\left(\begin{array}{c}\mathbb{T}_{RP}^k, \mathbb{T}_{IP}^k \in \mathbb{T}_{MS}, \\ \mathbb{F}_{RP}^k, \mathbb{F}_{IP}^k \in \mathbb{F}_{NMS}\end{array}\right)}$$

$$\left(\begin{array}{c}\left(1 - e^{-\left(\sum_{k=1}^q \left(\frac{1 +^\circ C\left(\mathfrak{Y}_{BF}^k\right)}{\sum_{k=1}^q \left(1 +^\circ C\left(\mathfrak{Y}_{BF}^k\right)\right)}\right)\left(-\ln\left(1 - \mathbb{T}_{RP}^k\right)\right)^\Xi\right)^{\frac{1}{\Xi}}} + \mathring{\mathbb{i}}\left(1 - e^{-\left(\sum_{k=1}^q \left(\frac{1 +^\circ C\left(\mathfrak{Y}_{BF}^k\right)}{\sum_{k=1}^q \left(1 +^\circ C\left(\mathfrak{Y}_{BF}^k\right)\right)}\right)\left(-\ln\left(1 - \mathbb{T}_{IP}^k\right)\right)^\Xi\right)^{\frac{1}{\Xi}}}\right)\right), \\ -\left(e^{-\left(\sum_{k=1}^q \left(\frac{1 +^\circ C\left(\mathfrak{Y}_{BF}^k\right)}{\sum_{k=1}^q \left(1 +^\circ C\left(\mathfrak{Y}_{BF}^k\right)\right)}\right)\left(-\ln\left(\left|\mathbb{F}_{RN}^k\right|\right)\right)^\Xi\right)^{\frac{1}{\Xi}}}\right) + \mathring{\mathbb{i}}\left(-\left(e^{-\left(\sum_{k=1}^q \left(\frac{1 +^\circ C\left(\mathfrak{Y}_{BF}^k\right)}{\sum_{k=1}^q \left(1 +^\circ C\left(\mathfrak{Y}_{BF}^k\right)\right)}\right)\left(-\ln\left(\left|\mathbb{F}_{IN}^k\right|\right)\right)^\Xi\right)^{\frac{1}{\Xi}}}\right)\right)\end{array}\right)$$

**Proof:** To consider the technique of mathematical induction, we derive that the above information is correct for all values of *q*. For this, we have *q* = 2, then

$$\left(\frac{1 +^\circ C\left(\mathfrak{Y}_{BF}^1\right)}{\sum_{k=1}^q \left(1 +^\circ C\left(\mathfrak{Y}_{BF}^k\right)\right)}\right)\mathfrak{Y}_{BF}^1 = \coprod_{\left(\begin{array}{c}\mathbb{T}_{RP}^1, \mathbb{T}_{IP}^1 \in \mathbb{T}_{MS}, \\ \mathbb{F}_{RP}^1, \mathbb{F}_{IP}^1 \in \mathbb{F}_{NMS}\end{array}\right)}$$

$$\left(\begin{array}{c}\left(1 - e^{-\left(\left(\frac{1 +^\circ C\left(\mathfrak{Y}_{BF}^1\right)}{\sum_{k=1}^q \left(1 +^\circ C\left(\mathfrak{Y}_{BF}^k\right)\right)}\right)\left(-\ln\left(1 - \mathbb{T}_{RP}^1\right)\right)^\Xi\right)^{\frac{1}{\Xi}}} + \mathring{\mathbb{i}}\left(1 - e^{-\left(\left(\frac{1 +^\circ C\left(\mathfrak{Y}_{BF}^1\right)}{\sum_{k=1}^q \left(1 +^\circ C\left(\mathfrak{Y}_{BF}^k\right)\right)}\right)\left(-\ln\left(1 - \mathbb{T}_{IP}^1\right)\right)^\Xi\right)^{\frac{1}{\Xi}}}\right)\right), \\ -\left(e^{-\left(\left(\frac{1 +^\circ C\left(\mathfrak{Y}_{BF}^1\right)}{\sum_{k=1}^q \left(1 +^\circ C\left(\mathfrak{Y}_{BF}^k\right)\right)}\right)\left(-\ln\left(\left|\mathbb{F}_{RN}^1\right|\right)\right)^\Xi\right)^{\frac{1}{\Xi}}}\right) + \mathring{\mathbb{i}}\left(-\left(e^{-\left(\left(\frac{1 +^\circ C\left(\mathfrak{Y}_{BF}^1\right)}{\sum_{k=1}^q \left(1 +^\circ C\left(\mathfrak{Y}_{BF}^k\right)\right)}\right)\left(-\ln\left(\left|\mathbb{F}_{IN}^1\right|\right)\right)^\Xi\right)^{\frac{1}{\Xi}}}\right)\right)\end{array}\right)$$

$$\left(\frac{1 +^\circ C\left(\mathfrak{Y}_{BF}^2\right)}{\sum_{k=1}^q \left(1 +^\circ C\left(\mathfrak{Y}_{BF}^k\right)\right)}\right)\mathfrak{Y}_{BF}^2 = \coprod_{\begin{pmatrix}\mathbb{T}_{RP}^2, \mathbb{T}_{IP}^2 \in \mathbb{T}_{MS},\\ \mathbb{F}_{RP}^2, \mathbb{F}_{IP}^2 \in \mathbb{F}_{NMS}\end{pmatrix}}$$

$$\left(\begin{pmatrix}1 - e^{-\left(\left(\left(\frac{1 +^\circ C\left(\mathfrak{Y}_{BF}^2\right)}{\sum_{k=1}^q \left(1 +^\circ C\left(\mathfrak{Y}_{BF}^k\right)\right)}\right)\left(-\ln\left(1-\mathbb{T}_{RP}^2\right)\right)^\Xi\right)^{\frac{1}{\Xi}}}} + \mathring{\imath}\left(1 - e^{-\left(\left(\left(\frac{1 +^\circ C\left(\mathfrak{Y}_{BF}^2\right)}{\sum_{k=1}^q \left(1 +^\circ C\left(\mathfrak{Y}_{BF}^k\right)\right)}\right)\left(-\ln\left(1-\mathbb{T}_{IP}^2\right)\right)^\Xi\right)^{\frac{1}{\Xi}}}}\right),\\ -\left(e^{-\left(\left(\left(\frac{1 +^\circ C\left(\mathfrak{Y}_{BF}^2\right)}{\sum_{k=1}^q \left(1 +^\circ C\left(\mathfrak{Y}_{BF}^k\right)\right)}\right)\left(-\ln\left(\left|\mathbb{F}_{RN}^2\right|\right)\right)^\Xi\right)^{\frac{1}{\Xi}}}}\right) + \mathring{\imath}\left(-\left(e^{-\left(\left(\left(\frac{1 +^\circ C\left(\mathfrak{Y}_{BF}^2\right)}{\sum_{k=1}^q \left(1 +^\circ C\left(\mathfrak{Y}_{BF}^k\right)\right)}\right)\left(-\ln\left(\left|\mathbb{F}_{IN}^2\right|\right)\right)^\Xi\right)^{\frac{1}{\Xi}}}}\right)\right)\end{pmatrix}\right)$$

Thus,

$$BCHFAAPA\left(\mathfrak{Y}_{BF}^1, \mathfrak{Y}_{BF}^2\right) = \left(\frac{1 +^\circ C\left(\mathfrak{Y}_{BF}^1\right)}{\sum_{k=1}^q \left(1 +^\circ C\left(\mathfrak{Y}_{BF}^k\right)\right)}\right)\mathfrak{Y}_{BF}^1 \oplus \left(\frac{1 +^\circ C\left(\mathfrak{Y}_{BF}^2\right)}{\sum_{k=1}^q \left(1 +^\circ C\left(\mathfrak{Y}_{BF}^k\right)\right)}\right)\mathfrak{Y}_{BF}^2$$

$$= \coprod_{\begin{pmatrix}\mathbb{T}_{RP}^1, \mathbb{T}_{IP}^1 \in \mathbb{T}_{MS},\\ \mathbb{F}_{RP}^1, \mathbb{F}_{IP}^1 \in \mathbb{F}_{NMS}\end{pmatrix}}$$

$$\left(\begin{pmatrix}1 - e^{-\left(\left(\left(\frac{1 +^\circ C\left(\mathfrak{Y}_{BF}^1\right)}{\sum_{k=1}^q \left(1 +^\circ C\left(\mathfrak{Y}_{BF}^k\right)\right)}\right)\left(-\ln\left(1-\mathbb{T}_{RP}^1\right)\right)^\Xi\right)^{\frac{1}{\Xi}}}} + \mathring{\imath}\left(1 - e^{-\left(\left(\left(\frac{1 +^\circ C\left(\mathfrak{Y}_{BF}^1\right)}{\sum_{k=1}^q \left(1 +^\circ C\left(\mathfrak{Y}_{BF}^k\right)\right)}\right)\left(-\ln\left(1-\mathbb{T}_{IP}^1\right)\right)^\Xi\right)^{\frac{1}{\Xi}}}}\right),\\ -\left(e^{-\left(\left(\left(\frac{1 +^\circ C\left(\mathfrak{Y}_{BF}^1\right)}{\sum_{k=1}^q \left(1 +^\circ C\left(\mathfrak{Y}_{BF}^k\right)\right)}\right)\left(-\ln\left(\left|\mathbb{F}_{RN}^1\right|\right)\right)^\Xi\right)^{\frac{1}{\Xi}}}}\right) + \mathring{\imath}\left(-\left(e^{-\left(\left(\left(\frac{1 +^\circ C\left(\mathfrak{Y}_{BF}^1\right)}{\sum_{k=1}^q \left(1 +^\circ C\left(\mathfrak{Y}_{BF}^k\right)\right)}\right)\left(-\ln\left(\left|\mathbb{F}_{IN}^1\right|\right)\right)^\Xi\right)^{\frac{1}{\Xi}}}}\right)\right)\end{pmatrix}\right)$$

$$\oplus \coprod_{\begin{pmatrix}\mathbb{T}_{RP}^2, \mathbb{T}_{IP}^2 \in \mathbb{T}_{MS},\\ \mathbb{F}_{RP}^2, \mathbb{F}_{IP}^2 \in \mathbb{F}_{NMS}\end{pmatrix}}$$

$$\left(\begin{pmatrix}1 - e^{-\left(\left(\left(\frac{1 +^\circ C\left(\mathfrak{Y}_{BF}^2\right)}{\sum_{k=1}^q \left(1 +^\circ C\left(\mathfrak{Y}_{BF}^k\right)\right)}\right)\left(-\ln\left(1-\mathbb{T}_{RP}^2\right)\right)^\Xi\right)^{\frac{1}{\Xi}}}} + \mathring{\imath}\left(1 - e^{-\left(\left(\left(\frac{1 +^\circ C\left(\mathfrak{Y}_{BF}^2\right)}{\sum_{k=1}^q \left(1 +^\circ C\left(\mathfrak{Y}_{BF}^k\right)\right)}\right)\left(-\ln\left(1-\mathbb{T}_{IP}^2\right)\right)^\Xi\right)^{\frac{1}{\Xi}}}}\right),\\ -\left(e^{-\left(\left(\left(\frac{1 +^\circ C\left(\mathfrak{Y}_{BF}^2\right)}{\sum_{k=1}^q \left(1 +^\circ C\left(\mathfrak{Y}_{BF}^k\right)\right)}\right)\left(-\ln\left(\left|\mathbb{F}_{RN}^2\right|\right)\right)^\Xi\right)^{\frac{1}{\Xi}}}}\right) + \mathring{\imath}\left(-\left(e^{-\left(\left(\left(\frac{1 +^\circ C\left(\mathfrak{Y}_{BF}^2\right)}{\sum_{k=1}^q \left(1 +^\circ C\left(\mathfrak{Y}_{BF}^k\right)\right)}\right)\left(-\ln\left(\left|\mathbb{F}_{IN}^2\right|\right)\right)^\Xi\right)^{\frac{1}{\Xi}}}}\right)\right)\end{pmatrix}\right)$$

$$= \coprod_{\begin{pmatrix} \mathbb{T}_{RP}^k, \mathbb{T}_{IP}^k \in \mathbb{T}_{MS}, \\ \mathbb{F}_{RP}^k, \mathbb{F}_{IP}^k \in \mathbb{F}_{NMS} \end{pmatrix}}$$

$$\left( \begin{pmatrix} 1 - e^{-\left( \sum_{k=1}^2 \left( \frac{1 +^\circ C\left( \mathfrak{Y}_{BF}^k \right)}{\sum_{k=1}^2 \left( 1 +^\circ C\left( \mathfrak{Y}_{BF}^k \right) \right)} \right) \left( -\ln\left( 1 - \mathbb{T}_{RP}^k \right) \right)^\Xi \right)^{\frac{1}{\Xi}}} + \mathring{\mathbb{i}} \left( 1 - e^{-\left( \sum_{k=1}^2 \left( \frac{1 +^\circ C\left( \mathfrak{Y}_{BF}^k \right)}{\sum_{k=1}^2 \left( 1 +^\circ C\left( \mathfrak{Y}_{BF}^k \right) \right)} \right) \left( -\ln\left( 1 - \mathbb{T}_{IP}^k \right) \right)^\Xi \right)^{\frac{1}{\Xi}}} \right), \\ -\left( e^{-\left( \sum_{k=1}^2 \left( \frac{1 +^\circ C\left( \mathfrak{Y}_{BF}^k \right)}{\sum_{k=1}^2 \left( 1 +^\circ C\left( \mathfrak{Y}_{BF}^k \right) \right)} \right) \left( -\ln\left( \left| \mathbb{F}_{RN}^k \right| \right) \right)^\Xi \right)^{\frac{1}{\Xi}}} \right) + \mathring{\mathbb{i}} \left( -\left( e^{-\left( \sum_{k=1}^2 \left( \frac{1 +^\circ C\left( \mathfrak{Y}_{BF}^k \right)}{\sum_{k=1}^2 \left( 1 +^\circ C\left( \mathfrak{Y}_{BF}^k \right) \right)} \right) \left( -\ln\left( \left| \mathbb{F}_{IN}^k \right| \right) \right)^\Xi \right)^{\frac{1}{\Xi}}} \right) \right) \end{pmatrix} \right)$$

Moreover, we consider that the proposed operator is held for $q = Q$, then

$$BCHFAAPA\left( \mathfrak{Y}_{BF}^1, \mathfrak{Y}_{BF}^2, \ldots, \mathfrak{Y}_{BF}^Q \right) = \coprod_{\begin{pmatrix} \mathbb{T}_{RP}^k, \mathbb{T}_{IP}^k \in \mathbb{T}_{MS}, \\ \mathbb{F}_{RP}^k, \mathbb{F}_{IP}^k \in \mathbb{F}_{NMS} \end{pmatrix}}$$

$$\left( \begin{pmatrix} 1 - e^{-\left( \sum_{k=1}^Q \left( \frac{1 +^\circ C\left( \mathfrak{Y}_{BF}^k \right)}{\sum_{k=1}^q \left( 1 +^\circ C\left( \mathfrak{Y}_{BF}^k \right) \right)} \right) \left( -\ln\left( 1 - \mathbb{T}_{RP}^k \right) \right)^\Xi \right)^{\frac{1}{\Xi}}} + \mathring{\mathbb{i}} \left( 1 - e^{-\left( \sum_{k=1}^Q \left( \frac{1 +^\circ C\left( \mathfrak{Y}_{BF}^k \right)}{\sum_{k=1}^q \left( 1 +^\circ C\left( \mathfrak{Y}_{BF}^k \right) \right)} \right) \left( -\ln\left( 1 - \mathbb{T}_{IP}^k \right) \right)^\Xi \right)^{\frac{1}{\Xi}}} \right), \\ -\left( e^{-\left( \sum_{k=1}^Q \left( \frac{1 +^\circ C\left( \mathfrak{Y}_{BF}^k \right)}{\sum_{k=1}^q \left( 1 +^\circ C\left( \mathfrak{Y}_{BF}^k \right) \right)} \right) \left( -\ln\left( \left| \mathbb{F}_{RN}^k \right| \right) \right)^\Xi \right)^{\frac{1}{\Xi}}} \right) + \mathring{\mathbb{i}} \left( -\left( e^{-\left( \sum_{k=1}^Q \left( \frac{1 +^\circ C\left( \mathfrak{Y}_{BF}^k \right)}{\sum_{k=1}^q \left( 1 +^\circ C\left( \mathfrak{Y}_{BF}^k \right) \right)} \right) \left( -\ln\left( \left| \mathbb{F}_{IN}^k \right| \right) \right)^\Xi \right)^{\frac{1}{\Xi}}} \right) \right) \end{pmatrix} \right)$$

Thus, we evaluate it for $q = Q + 1$, such as

$$BCHFAAPA\left( \mathfrak{Y}_{BF}^1, \mathfrak{Y}_{BF}^2, \ldots, \mathfrak{Y}_{BF}^{Q+1} \right) = \left( \frac{1 +^\circ C\left( \mathfrak{Y}_{BF}^1 \right)}{\sum_{k=1}^{Q+1} \left( 1 +^\circ C\left( \mathfrak{Y}_{BF}^k \right) \right)} \right) \mathfrak{Y}_{BF}^1 \oplus \left( \frac{1 +^\circ C\left( \mathfrak{Y}_{BF}^2 \right)}{\sum_{k=1}^{Q+1} \left( 1 +^\circ C\left( \mathfrak{Y}_{BF}^k \right) \right)} \right) \mathfrak{Y}_{BF}^2$$

$$\oplus \ldots \oplus \left( \frac{1 +^\circ C\left( \mathfrak{Y}_{BF}^Q \right)}{\sum_{k=1}^{Q+1} \left( 1 +^\circ C\left( \mathfrak{Y}_{BF}^k \right) \right)} \right) \mathfrak{Y}_{BF}^Q \oplus \left( \frac{1 +^\circ C\left( \mathfrak{Y}_{BF}^{Q+1} \right)}{\sum_{k=1}^{Q+1} \left( 1 +^\circ C\left( \mathfrak{Y}_{BF}^k \right) \right)} \right) \mathfrak{Y}_{BF}^{Q+1}$$

$$= \sum_{k=1}^Q \left( \frac{1 +^\circ C\left( \mathfrak{Y}_{BF}^k \right)}{\sum_{k=1}^q \left( 1 +^\circ C\left( \mathfrak{Y}_{BF}^k \right) \right)} \right) \mathfrak{Y}_{BF}^k \oplus \left( \frac{1 +^\circ C\left( \mathfrak{Y}_{BF}^{Q+1} \right)}{\sum_{k=1}^{Q+1} \left( 1 +^\circ C\left( \mathfrak{Y}_{BF}^k \right) \right)} \right) \mathfrak{Y}_{BF}^{Q+1}$$

$$= \coprod_{\left(\begin{array}{c} \mathbb{T}_{RP}^k, \mathbb{T}_{IP}^k \in \mathbb{T}_{MS}, \\ \mathbb{F}_{RP}^k, \mathbb{F}_{IP}^k \in \mathbb{F}_{NMS} \end{array}\right)}$$

$$\left(\begin{array}{c} \left(1 - e^{-\left(\sum_{k=1}^{Q}\left(\frac{1 + ^\circ C\left(\mathfrak{Y}_{BF}^k\right)}{\sum_{k=1}^{q}\left(1 + ^\circ C\left(\mathfrak{Y}_{BF}^k\right)\right)}\right)\left(-\ln\left(1 - \mathbb{T}_{RP}^k\right)\right)^\Xi\right)^{\frac{1}{\Xi}}} + \mathring{\text{i}}\left(1 - e^{-\left(\sum_{k=1}^{Q}\left(\frac{1 + ^\circ C\left(\mathfrak{Y}_{BF}^k\right)}{\sum_{k=1}^{q}\left(1 + ^\circ C\left(\mathfrak{Y}_{BF}^k\right)\right)}\right)\left(-\ln\left(1 - \mathbb{T}_{IP}^k\right)\right)^\Xi\right)^{\frac{1}{\Xi}}}\right), \\ -\left(e^{-\left(\sum_{k=1}^{Q}\left(\frac{1 + ^\circ C\left(\mathfrak{Y}_{BF}^k\right)}{\sum_{k=1}^{q}\left(1 + ^\circ C\left(\mathfrak{Y}_{BF}^k\right)\right)}\right)\left(-\ln\left(\left|\mathbb{F}_{RN}^k\right|\right)\right)^\Xi\right)^{\frac{1}{\Xi}}}\right) + \mathring{\text{i}}\left(-\left(e^{-\left(\sum_{k=1}^{Q}\left(\frac{1 + ^\circ C\left(\mathfrak{Y}_{BF}^k\right)}{\sum_{k=1}^{q}\left(1 + ^\circ C\left(\mathfrak{Y}_{BF}^k\right)\right)}\right)\left(-\ln\left(\left|\mathbb{F}_{IN}^k\right|\right)\right)^\Xi\right)^{\frac{1}{\Xi}}}\right)\right) \end{array}\right)$$

$$\oplus \left(\frac{1 + ^\circ C\left(\mathfrak{Y}_{BF}^{Q+1}\right)}{\sum_{k=1}^{Q+1}\left(1 + ^\circ C\left(\mathfrak{Y}_{BF}^k\right)\right)}\right)\mathfrak{Y}_{BF}^{Q+1}$$

$$= \coprod_{\left(\begin{array}{c} \mathbb{T}_{RP}^k, \mathbb{T}_{IP}^k \in \mathbb{T}_{MS}, \\ \mathbb{F}_{RP}^k, \mathbb{F}_{IP}^k \in \mathbb{F}_{NMS} \end{array}\right)}$$

$$\left(\begin{array}{c} \left(1 - e^{-\left(\sum_{k=1}^{Q}\left(\frac{1 + ^\circ C\left(\mathfrak{Y}_{BF}^k\right)}{\sum_{k=1}^{q}\left(1 + ^\circ C\left(\mathfrak{Y}_{BF}^k\right)\right)}\right)\left(-\ln\left(1 - \mathbb{T}_{RP}^k\right)\right)^\Xi\right)^{\frac{1}{\Xi}}} + \mathring{\text{i}}\left(1 - e^{-\left(\sum_{k=1}^{Q}\left(\frac{1 + ^\circ C\left(\mathfrak{Y}_{BF}^k\right)}{\sum_{k=1}^{q}\left(1 + ^\circ C\left(\mathfrak{Y}_{BF}^k\right)\right)}\right)\left(-\ln\left(1 - \mathbb{T}_{IP}^k\right)\right)^\Xi\right)^{\frac{1}{\Xi}}}\right), \\ -\left(e^{-\left(\sum_{k=1}^{Q}\left(\frac{1 + ^\circ C\left(\mathfrak{Y}_{BF}^k\right)}{\sum_{k=1}^{q}\left(1 + ^\circ C\left(\mathfrak{Y}_{BF}^k\right)\right)}\right)\left(-\ln\left(\left|\mathbb{F}_{RN}^k\right|\right)\right)^\Xi\right)^{\frac{1}{\Xi}}}\right) + \mathring{\text{i}}\left(-\left(e^{-\left(\sum_{k=1}^{Q}\left(\frac{1 + ^\circ C\left(\mathfrak{Y}_{BF}^k\right)}{\sum_{k=1}^{q}\left(1 + ^\circ C\left(\mathfrak{Y}_{BF}^k\right)\right)}\right)\left(-\ln\left(\left|\mathbb{F}_{IN}^k\right|\right)\right)^\Xi\right)^{\frac{1}{\Xi}}}\right)\right) \end{array}\right)$$

$$\oplus \coprod_{\left(\begin{array}{c} \mathbb{T}_{RP}^{Q+1}, \mathbb{T}_{IP}^{Q+1} \in \mathbb{T}_{MS}, \\ \mathbb{F}_{RP}^{Q+1}, \mathbb{F}_{IP}^{Q+1} \in \mathbb{F}_{NMS} \end{array}\right)}$$

$$\left(\begin{array}{c} \left(1 - e^{-\left(\left(\frac{1 + ^\circ C\left(\mathfrak{Y}_{BF}^{Q+1}\right)}{\sum_{k=1}^{Q+1}\left(1 + ^\circ C\left(\mathfrak{Y}_{BF}^k\right)\right)}\right)\left(-\ln\left(1 - \mathbb{T}_{RP}^{Q+1}\right)\right)^\Xi\right)^{\frac{1}{\Xi}}} + \mathring{\text{i}}\left(1 - e^{-\left(\left(\frac{1 + ^\circ C\left(\mathfrak{Y}_{BF}^{Q+1}\right)}{\sum_{k=1}^{Q+1}\left(1 + ^\circ C\left(\mathfrak{Y}_{BF}^k\right)\right)}\right)\left(-\ln\left(1 - \mathbb{T}_{IP}^{Q+1}\right)\right)^\Xi\right)^{\frac{1}{\Xi}}}\right), \\ -\left(e^{-\left(\left(\frac{1 + ^\circ C\left(\mathfrak{Y}_{BF}^{Q+1}\right)}{\sum_{k=1}^{Q+1}\left(1 + ^\circ C\left(\mathfrak{Y}_{BF}^k\right)\right)}\right)\left(-\ln\left(\left|\mathbb{F}_{RN}^{Q+1}\right|\right)\right)^\Xi\right)^{\frac{1}{\Xi}}}\right) + \mathring{\text{i}}\left(-\left(e^{-\left(\left(\frac{1 + ^\circ C\left(\mathfrak{Y}_{BF}^{Q+1}\right)}{\sum_{k=1}^{Q+1}\left(1 + ^\circ C\left(\mathfrak{Y}_{BF}^k\right)\right)}\right)\left(-\ln\left(\left|\mathbb{F}_{IN}^{Q+1}\right|\right)\right)^\Xi\right)^{\frac{1}{\Xi}}}\right)\right) \end{array}\right)$$

$$= \prod_{\left(\begin{array}{c} \mathbb{T}_{RP}^k, \mathbb{T}_{IP}^k \in \mathbb{T}_{MS}, \\ \mathbb{F}_{RP}^k, \mathbb{F}_{IP}^k \in \mathbb{F}_{NMS} \end{array}\right)}$$

$$\left( \left( 1 - e^{-\left( \sum_{k=1}^{Q+1} \left( \frac{1 +^\circ \mathrm{C}\left(\mathfrak{Y}_{BF}^k\right)}{\sum_{k=1}^q \left(1 +^\circ \mathrm{C}\left(\mathfrak{Y}_{BF}^k\right)\right)} \right) \left(-\ln\left(1-\mathbb{T}_{RP}^k\right)\right)^\Xi \right)^{\frac{1}{\Xi}}} + \mathring{\mathbb{i}} \left( 1 - e^{-\left( \sum_{k=1}^{Q+1} \left( \frac{1 +^\circ \mathrm{C}\left(\mathfrak{Y}_{BF}^k\right)}{\sum_{k=1}^q \left(1 +^\circ \mathrm{C}\left(\mathfrak{Y}_{BF}^k\right)\right)} \right) \left(-\ln\left(1-\mathbb{T}_{IP}^k\right)\right)^\Xi \right)^{\frac{1}{\Xi}}} \right) \right),$$
$$\left( -\left( e^{-\left( \sum_{k=1}^{Q+!} \left( \frac{1 +^\circ \mathrm{C}\left(\mathfrak{Y}_{BF}^k\right)}{\sum_{k=1}^q \left(1 +^\circ \mathrm{C}\left(\mathfrak{Y}_{BF}^k\right)\right)} \right) \left(-\ln\left(\left|\mathbb{F}_{RN}^k\right|\right)\right)^\Xi \right)^{\frac{1}{\Xi}}} \right) + \mathring{\mathbb{i}} \left( -\left( e^{-\left( \sum_{k=1}^{Q+!} \left( \frac{1 +^\circ \mathrm{C}\left(\mathfrak{Y}_{BF}^k\right)}{\sum_{k=1}^q \left(1 +^\circ \mathrm{C}\left(\mathfrak{Y}_{BF}^k\right)\right)} \right) \left(-\ln\left(\left|\mathbb{F}_{IN}^k\right|\right)\right)^\Xi \right)^{\frac{1}{\Xi}}} \right) \right) \right)$$

Hence, our proposed technique is held for all positive integers. Further, we simplify the idempotency, monotonicity, and boundedness of the proposed techniques.

**Proposition 1:** For any collection of BCHFNs. Then

1) When $\mathfrak{Y}_{BF}^k = \mathfrak{Y}_{BF}$, thus

$$BCHFAAPA\left(\mathfrak{Y}_{BF}^1, \mathfrak{Y}_{BF}^2, \ldots, \mathfrak{Y}_{BF}^q\right) = \mathfrak{Y}_{BF}.$$

2) When $\mathfrak{Y}_{BF}^k \leq \mathfrak{Y}_{BF}^{\#k}$, thus

$$BCHFAAPA\left(\mathfrak{Y}_{BF}^1, \mathfrak{Y}_{BF}^2, \ldots, \mathfrak{Y}_{BF}^q\right) \leq BCHFAAPA\left(\mathfrak{Y}_{BF}^{\#1}, \mathfrak{Y}_{BF}^{\#2}, \ldots, \mathfrak{Y}_{BF}^{\#q}\right).$$

3) When $\mathfrak{Y}_{BF}^- = min\left(\mathfrak{Y}_{BF}^1, \mathfrak{Y}_{BF}^2, \ldots, \mathfrak{Y}_{BF}^q\right)$ and $\mathfrak{Y}_{BF}^+ = max\left(\mathfrak{Y}_{BF}^1, \mathfrak{Y}_{BF}^2, \ldots, \mathfrak{Y}_{BF}^q\right)$, thus

$$\mathfrak{Y}_{BF}^- \leq BCHFAAPA\left(\mathfrak{Y}_{BF}^1, \mathfrak{Y}_{BF}^2, \ldots, \mathfrak{Y}_{BF}^q\right) \leq \mathfrak{Y}_{BF}^+.$$

**Proof:** Omitted.

**Definition 10:** For any collection of BCHFNs, the BCHFAAPWA operator is presented and arranged in the shape:

$$BCHFAAPWA\left(\mathfrak{Y}_{BF}^1, \mathfrak{Y}_{BF}^2, \ldots, \mathfrak{Y}_{BF}^q\right) = \left( \frac{{}^\circ\mathrm{F}_{wv}^1 \left(1 +^\circ \mathrm{C}\left(\mathfrak{Y}_{BF}^1\right)\right)}{\sum_{k=1}^q {}^\circ\mathrm{F}_{wv}^k \left(1 +^\circ \mathrm{C}\left(\mathfrak{Y}_{BF}^k\right)\right)} \right) \mathfrak{Y}_{BF}^1$$

$$\oplus \left( \frac{{}^\circ\mathrm{F}_{wv}^2 \left(1 +^\circ \mathrm{C}\left(\mathfrak{Y}_{BF}^2\right)\right)}{\sum_{k=1}^q {}^\circ\mathrm{F}_{wv}^k \left(1 +^\circ \mathrm{C}\left(\mathfrak{Y}_{BF}^k\right)\right)} \right) \mathfrak{Y}_{BF}^2 \oplus \ldots \oplus \left( \frac{{}^\circ\mathrm{F}_{wv}^q \left(1 +^\circ \mathrm{C}\left(\mathfrak{Y}_{BF}^q\right)\right)}{\sum_{k=1}^q {}^\circ\mathrm{F}_{wv}^k \left(1 +^\circ \mathrm{C}\left(\mathfrak{Y}_{BF}^k\right)\right)} \right) \mathfrak{Y}_{BF}^q$$

$$= \sum_{k=1}^q \left( \frac{{}^\circ\mathrm{F}_{wv}^k \left(1 +^\circ \mathrm{C}\left(\mathfrak{Y}_{BF}^k\right)\right)}{\sum_{k=1}^q {}^\circ\mathrm{F}_{wv}^k \left(1 +^\circ \mathrm{C}\left(\mathfrak{Y}_{BF}^k\right)\right)} \right) \mathfrak{Y}_{BF}^k$$

Where, ${}^\circ\mathrm{F}_{wv}^k \in [0, 1]$ with $\sum_{k=1}^q {}^\circ\mathrm{F}_{wv}^k = 1$ represents the weight vectors.

**Theorem 2:** For any collection of BCHFNs, then we prove that the aggregated value of the above information is again a BCHFN, such as

$$
BCHFAAPWA\left(\mathfrak{Y}^1_{BF}, \mathfrak{Y}^2_{BF}, \ldots, \mathfrak{Y}^q_{BF}\right) = \coprod_{\left(\begin{array}{c}\mathbb{T}^k_{RP}, \mathbb{T}^k_{IP} \in \mathbb{T}_{MS}, \\ \mathbb{F}^k_{RP}, \mathbb{F}^k_{IP} \in \mathbb{F}_{NMS}\end{array}\right)}
$$

$$
\left(\begin{array}{c}
1 - e^{-\left(\sum_{k=1}^q \left(\frac{{}^\circ\mathbb{F}^k_{wv}\left(1 +{}^\circ C\left(\mathfrak{Y}^k_{BF}\right)\right)}{\sum_{k=1}^q {}^\circ\mathbb{F}^k_{wv}\left(1 +{}^\circ C\left(\mathfrak{Y}^k_{BF}\right)\right)}\right)\left(-\ln\left(1-\mathbb{T}^k_{RP}\right)\right)^\Xi\right)^{\frac{1}{\Xi}}} + \\[4mm]
\mathring{\mathbb{i}}\left(1 - e^{-\left(\sum_{k=1}^q \left(\frac{{}^\circ\mathbb{F}^k_{wv}\left(1 +{}^\circ C\left(\mathfrak{Y}^k_{BF}\right)\right)}{\sum_{k=1}^q {}^\circ\mathbb{F}^k_{wv}\left(1 +{}^\circ C\left(\mathfrak{Y}^k_{BF}\right)\right)}\right)\left(-\ln\left(1-\mathbb{T}^k_{IP}\right)\right)^\Xi\right)^{\frac{1}{\Xi}}}\right), \\[6mm]
-\left(e^{-\left(\sum_{k=1}^q \left(\frac{{}^\circ\mathbb{F}^k_{wv}\left(1 +{}^\circ C\left(\mathfrak{Y}^k_{BF}\right)\right)}{\sum_{k=1}^q {}^\circ\mathbb{F}^k_{wv}\left(1 +{}^\circ C\left(\mathfrak{Y}^k_{BF}\right)\right)}\right)\left(-\ln\left(\left|\mathbb{F}^k_{RN}\right|\right)\right)^\Xi\right)^{\frac{1}{\Xi}}}\right) + \\[6mm]
\mathring{\mathbb{i}}\left(-\left(e^{-\left(\sum_{k=1}^q \left(\frac{{}^\circ\mathbb{F}^k_{wv}\left(1 +{}^\circ C\left(\mathfrak{Y}^k_{BF}\right)\right)}{\sum_{k=1}^q {}^\circ\mathbb{F}^k_{wv}\left(1 +{}^\circ C\left(\mathfrak{Y}^k_{BF}\right)\right)}\right)\left(-\ln\left(\left|\mathbb{F}^k_{IN}\right|\right)\right)^\Xi\right)^{\frac{1}{\Xi}}}\right)\right)
\end{array}\right)
$$

Further, we simplify the idempotency, monotonicity, and boundedness of the proposed techniques.

**Proposition 2:** For any collection of BCHFNs. Then

1) When $\mathfrak{Y}^k_{BF} = \mathfrak{Y}_{BF}$, thus

$$
BCHFAAPWA\left(\mathfrak{Y}^1_{BF}, \mathfrak{Y}^2_{BF}, \ldots, \mathfrak{Y}^q_{BF}\right) = \mathfrak{Y}_{BF}.
$$

2) When $\mathfrak{Y}^k_{BF} \le \mathfrak{Y}^{\#k}_{BF}$, thus

$$
BCHFAAPWA\left(\mathfrak{Y}^1_{BF}, \mathfrak{Y}^2_{BF}, \ldots, \mathfrak{Y}^q_{BF}\right) \le BCHFAAPWA\left(\mathfrak{Y}^{\#1}_{BF}, \mathfrak{Y}^{\#2}_{BF}, \ldots, \mathfrak{Y}^{\#q}_{BF}\right).
$$

3) When $\mathfrak{Y}^-_{BF} = \min\left(\mathfrak{Y}^1_{BF}, \mathfrak{Y}^2_{BF}, \ldots, \mathfrak{Y}^q_{BF}\right)$ and $\mathfrak{Y}^+_{BF} = \max\left(\mathfrak{Y}^1_{BF}, \mathfrak{Y}^2_{BF}, \ldots, \mathfrak{Y}^q_{BF}\right)$, thus

$$
\mathfrak{Y}^-_{BF} \le BCHFAAPWA\left(\mathfrak{Y}^1_{BF}, \mathfrak{Y}^2_{BF}, \ldots, \mathfrak{Y}^q_{BF}\right) \le \mathfrak{Y}^+_{BF}.
$$

**Proof:** Omitted.

**Definition 11:** For any collection of BCHFNs, the BCHFAAPG operator is presented and arranged in the shape:

$$BCHFAAPG\left(\mathfrak{Y}_{BF}^1, \mathfrak{Y}_{BF}^2, \ldots, \mathfrak{Y}_{BF}^q\right) = \mathfrak{Y}_{BF}^1 \left(\frac{1 +^\circ C\left(\mathfrak{Y}_{BF}^1\right)}{\sum_{k=1}^q \left(1 +^\circ C\left(\mathfrak{Y}_{BF}^k\right)\right)}\right)$$

$$\otimes \mathfrak{Y}_{BF}^2 \left(\frac{1 +^\circ C\left(\mathfrak{Y}_{BF}^2\right)}{\sum_{k=1}^q \left(1 +^\circ C\left(\mathfrak{Y}_{BF}^k\right)\right)}\right) \otimes \ldots \otimes \mathfrak{Y}_{BF}^q \left(\frac{1 +^\circ C\left(\mathfrak{Y}_{BF}^q\right)}{\sum_{k=1}^q \left(1 +^\circ C\left(\mathfrak{Y}_{BF}^k\right)\right)}\right)$$

$$= \prod_{k=1}^q \mathfrak{Y}_{BF}^k \left(\frac{1 +^\circ C\left(\mathfrak{Y}_{BF}^k\right)}{\sum_{k=1}^q \left(1 +^\circ C\left(\mathfrak{Y}_{BF}^k\right)\right)}\right)$$

**Theorem 3:** For any collection of BCHFNs, then we prove that the aggregated value of the above information is again a BCHFN, such as

$$BCHFAAPG\left(\mathfrak{Y}_{BF}^1, \mathfrak{Y}_{BF}^2, \ldots, \mathfrak{Y}_{BF}^q\right) = \coprod_{\left(\begin{array}{c}\mathbb{T}_{RP}^k, \mathbb{T}_{IP}^k \in \mathbb{T}_{MS}, \\ \mathbb{F}_{RP}^k, \mathbb{F}_{IP}^k \in \mathbb{F}_{NMS}\end{array}\right)}$$

$$\left(\left(e^{-\left(\sum_{k=1}^q \left(\frac{1 +^\circ C\left(\mathfrak{Y}_{BF}^k\right)}{\sum_{k=1}^q \left(1 +^\circ C\left(\mathfrak{Y}_{BF}^k\right)\right)}\right)\left(-\ln\left(\mathbb{T}_{RP}^k\right)\right)^\Xi\right)^{\frac{1}{\Xi}}} + \mathring{\mathbb{i}}\left(e^{-\left(\sum_{k=1}^q \left(\frac{1 +^\circ C\left(\mathfrak{Y}_{BF}^k\right)}{\sum_{k=1}^q \left(1 +^\circ C\left(\mathfrak{Y}_{BF}^k\right)\right)}\right)\left(-\ln\left(\mathbb{T}_{IP}^k\right)\right)^\Xi\right)^{\frac{1}{\Xi}}}\right)\right),\right.$$

$$-1 + \left(e^{-\left(\sum_{k=1}^q \left(\frac{1 +^\circ C\left(\mathfrak{Y}_{BF}^k\right)}{\sum_{k=1}^q \left(1 +^\circ C\left(\mathfrak{Y}_{BF}^k\right)\right)}\right)\left(-\ln\left(\left(1 +\mathbb{F}_{RN}^k\right)\right)\right)^\Xi\right)^{\frac{1}{\Xi}}}\right) +$$

$$\left.\mathring{\mathbb{i}}\left(-1 + \left(e^{-\left(\sum_{k=1}^q \left(\frac{1 +^\circ C\left(\mathfrak{Y}_{BF}^k\right)}{\sum_{k=1}^q \left(1 +^\circ C\left(\mathfrak{Y}_{BF}^k\right)\right)}\right)\left(-\ln\left(\left(1 +\mathbb{F}_{IN}^k\right)\right)\right)^\Xi\right)^{\frac{1}{\Xi}}}\right)\right)\right)$$

**Proof:** To consider the technique of mathematical induction, we derive that the above information is correct for all values of $q$. For this, we have $q = 2$, then

$$\mathfrak{Y}^1_{BF}\left(\frac{1 +^\circ C(\mathfrak{Y}^1_{BF})}{\sum_{k=1}^q \left(1 +^\circ C(\mathfrak{Y}^k_{BF})\right)}\right) = \coprod_{\left(\begin{array}{c}\mathbb{T}^1_{RP}, \mathbb{T}^1_{IP} \in \mathbb{T}_{MS},\\ \mathbb{F}^1_{RP}, \mathbb{F}^1_{IP} \in \mathbb{F}_{NMS}\end{array}\right)}$$

$$\left(\left(e^{-\left(\left(\left(\frac{1 +^\circ C(\mathfrak{Y}^1_{BF})}{\sum_{k=1}^q \left(1 +^\circ C(\mathfrak{Y}^k_{BF})\right)}\right)\left(-\ln\left(\mathbb{T}^1_{RP}\right)\right)^\Xi\right)^{\frac{1}{\Xi}}} + \mathring{\imath}\left(e^{-\left(\left(\left(\frac{1 +^\circ C(\mathfrak{Y}^1_{BF})}{\sum_{k=1}^q \left(1 +^\circ C(\mathfrak{Y}^k_{BF})\right)}\right)\left(-\ln\left(\mathbb{T}^1_{IP}\right)\right)^\Xi\right)^{\frac{1}{\Xi}}}\right)\right),\right.$$

$$-1 + \left(e^{-\left(\left(\left(\frac{1 +^\circ C(\mathfrak{Y}^1_{BF})}{\sum_{k=1}^q \left(1 +^\circ C(\mathfrak{Y}^k_{BF})\right)}\right)\left(-\ln\left(\left(1+\mathbb{F}^1_{RN}\right)\right)\right)^\Xi\right)^{\frac{1}{\Xi}}}\right) +$$

$$\left.\mathring{\imath}\left(-1 + \left(e^{-\left(\left(\left(\frac{1 +^\circ C(\mathfrak{Y}^1_{BF})}{\sum_{k=1}^q \left(1 +^\circ C(\mathfrak{Y}^k_{BF})\right)}\right)\left(-\ln\left(\left(1+\mathbb{F}^1_{IN}\right)\right)\right)^\Xi\right)^{\frac{1}{\Xi}}}\right)\right)\right)$$

$$\mathfrak{Y}^2_{BF}\left(\frac{1 +^\circ C(\mathfrak{Y}^2_{BF})}{\sum_{k=1}^q \left(1 +^\circ C(\mathfrak{Y}^k_{BF})\right)}\right) = \coprod_{\left(\begin{array}{c}\mathbb{T}^2_{RP}, \mathbb{T}^2_{IP} \in \mathbb{T}_{MS},\\ \mathbb{F}^2_{RP}, \mathbb{F}^2_{IP} \in \mathbb{F}_{NMS}\end{array}\right)}$$

$$\left(\left(e^{-\left(\left(\left(\frac{1 +^\circ C(\mathfrak{Y}^2_{BF})}{\sum_{k=1}^q \left(1 +^\circ C(\mathfrak{Y}^k_{BF})\right)}\right)\left(-\ln\left(\mathbb{T}^2_{RP}\right)\right)^\Xi\right)^{\frac{1}{\Xi}}} + \mathring{\imath}\left(e^{-\left(\left(\left(\frac{1 +^\circ C(\mathfrak{Y}^2_{BF})}{\sum_{k=1}^q \left(1 +^\circ C(\mathfrak{Y}^k_{BF})\right)}\right)\left(-\ln\left(\mathbb{T}^2_{IP}\right)\right)^\Xi\right)^{\frac{1}{\Xi}}}\right)\right),\right.$$

$$-1 + \left(e^{-\left(\left(\left(\frac{1 +^\circ C(\mathfrak{Y}^2_{BF})}{\sum_{k=1}^q \left(1 +^\circ C(\mathfrak{Y}^k_{BF})\right)}\right)\left(-\ln\left(\left(1+\mathbb{F}^2_{RN}\right)\right)\right)^\Xi\right)^{\frac{1}{\Xi}}}\right) +$$

$$\left.\mathring{\imath}\left(-1 + \left(e^{-\left(\left(\left(\frac{1 +^\circ C(\mathfrak{Y}^2_{BF})}{\sum_{k=1}^q \left(1 +^\circ C(\mathfrak{Y}^k_{BF})\right)}\right)\left(-\ln\left(\left(1+\mathbb{F}^2_{IN}\right)\right)\right)^\Xi\right)^{\frac{1}{\Xi}}}\right)\right)\right)$$

Thus,

$$BCHFAAPG\left(\mathfrak{Y}_{BF}^1, \mathfrak{Y}_{BF}^2\right) = \mathfrak{Y}_{BF}^1 \left(\frac{1+{}^\circ C\left(\mathfrak{Y}_{BF}^1\right)}{\sum_{k=1}^q \left(1+{}^\circ C\left(\mathfrak{Y}_{BF}^k\right)\right)}\right) \otimes \mathfrak{Y}_{BF}^2 \left(\frac{1+{}^\circ C\left(\mathfrak{Y}_{BF}^2\right)}{\sum_{k=1}^q \left(1+{}^\circ C\left(\mathfrak{Y}_{BF}^k\right)\right)}\right)$$

$$= \coprod_{\begin{pmatrix} \mathbb{T}_{RP}^1, \mathbb{T}_{IP}^1 \in \mathbb{T}_{MS}, \\ \mathbb{F}_{RP}^1, \mathbb{F}_{IP}^1 \in \mathbb{F}_{NMS} \end{pmatrix}}$$

$$\left(\begin{array}{c} \left(e^{-\left(\left(\left(\frac{1+{}^\circ C\left(\mathfrak{Y}_{BF}^1\right)}{\sum_{k=1}^q \left(1+{}^\circ C\left(\mathfrak{Y}_{BF}^k\right)\right)}\right)\left(-\ln\left(\mathbb{T}_{RP}^1\right)\right)^\Xi\right)^{\frac{1}{\Xi}}} + \mathring{\mathbb{i}}\left(e^{-\left(\left(\left(\frac{1+{}^\circ C\left(\mathfrak{Y}_{BF}^1\right)}{\sum_{k=1}^q \left(1+{}^\circ C\left(\mathfrak{Y}_{BF}^k\right)\right)}\right)\left(-\ln\left(\mathbb{T}_{IP}^1\right)\right)^\Xi\right)^{\frac{1}{\Xi}}}\right)\right), \\ -1 + \left(e^{-\left(\left(\left(\frac{1+{}^\circ C\left(\mathfrak{Y}_{BF}^1\right)}{\sum_{k=1}^q \left(1+{}^\circ C\left(\mathfrak{Y}_{BF}^k\right)\right)}\right)\left(-\ln\left(\left(1+\mathbb{F}_{RN}^1\right)\right)\right)^\Xi\right)^{\frac{1}{\Xi}}}\right) + \\ \mathring{\mathbb{i}}\left(-1 + \left(e^{-\left(\left(\left(\frac{1+{}^\circ C\left(\mathfrak{Y}_{BF}^1\right)}{\sum_{k=1}^q \left(1+{}^\circ C\left(\mathfrak{Y}_{BF}^k\right)\right)}\right)\left(-\ln\left(\left(1+\mathbb{F}_{IN}^1\right)\right)\right)^\Xi\right)^{\frac{1}{\Xi}}}\right)\right) \end{array}\right)$$

$$\otimes \coprod_{\begin{pmatrix} \mathbb{T}_{RP}^2, \mathbb{T}_{IP}^2 \in \mathbb{T}_{MS}, \\ \mathbb{F}_{RP}^2, \mathbb{F}_{IP}^2 \in \mathbb{F}_{NMS} \end{pmatrix}}$$

$$\left(\begin{array}{c} \left(e^{-\left(\left(\left(\frac{1+{}^\circ C\left(\mathfrak{Y}_{BF}^2\right)}{\sum_{k=1}^q \left(1+{}^\circ C\left(\mathfrak{Y}_{BF}^k\right)\right)}\right)\left(-\ln\left(\mathbb{T}_{RP}^2\right)\right)^\Xi\right)^{\frac{1}{\Xi}}} + \mathring{\mathbb{i}}\left(e^{-\left(\left(\left(\frac{1+{}^\circ C\left(\mathfrak{Y}_{BF}^2\right)}{\sum_{k=1}^q \left(1+{}^\circ C\left(\mathfrak{Y}_{BF}^k\right)\right)}\right)\left(-\ln\left(\mathbb{T}_{IP}^2\right)\right)^\Xi\right)^{\frac{1}{\Xi}}}\right)\right), \\ -1 + \left(e^{-\left(\left(\left(\frac{1+{}^\circ C\left(\mathfrak{Y}_{BF}^2\right)}{\sum_{k=1}^q \left(1+{}^\circ C\left(\mathfrak{Y}_{BF}^k\right)\right)}\right)\left(-\ln\left(\left(1+\mathbb{F}_{RN}^2\right)\right)\right)^\Xi\right)^{\frac{1}{\Xi}}}\right) + \\ \mathring{\mathbb{i}}\left(-1 + \left(e^{-\left(\left(\left(\frac{1+{}^\circ C\left(\mathfrak{Y}_{BF}^2\right)}{\sum_{k=1}^q \left(1+{}^\circ C\left(\mathfrak{Y}_{BF}^k\right)\right)}\right)\left(-\ln\left(\left(1+\mathbb{F}_{IN}^2\right)\right)\right)^\Xi\right)^{\frac{1}{\Xi}}}\right)\right) \end{array}\right)$$

$$= \coprod_{\left( \begin{array}{c} \mathbb{T}^k_{RP}, \mathbb{T}^k_{IP} \in \mathbb{T}_{MS}, \\ \mathbb{F}^k_{RP}, \mathbb{F}^k_{IP} \in \mathbb{F}_{NMS} \end{array} \right)}$$

$$\left( \begin{array}{c} \left( e^{-\left( \sum_{k=1}^2 \left( \frac{1 +^\circ C(\mathfrak{Y}^k_{BF})}{\sum_{k=1}^2 \left(1 +^\circ C(\mathfrak{Y}^k_{BF})\right)} \right) \left(-\ln\left(\mathbb{T}^k_{RP}\right)\right)^\Xi \right)^{\frac{1}{\Xi}}} + \mathring{\mathfrak{i}} \left( e^{-\left( \sum_{k=1}^2 \left( \frac{1 +^\circ C(\mathfrak{Y}^k_{BF})}{\sum_{k=1}^2 \left(1 +^\circ C(\mathfrak{Y}^k_{BF})\right)} \right) \left(-\ln\left(\mathbb{T}^k_{IP}\right)\right)^\Xi \right)^{\frac{1}{\Xi}}} \right), \\ \\ -1 + \left( e^{-\left( \sum_{k=1}^2 \left( \frac{1 +^\circ C(\mathfrak{Y}^k_{BF})}{\sum_{k=1}^2 \left(1 +^\circ C(\mathfrak{Y}^k_{BF})\right)} \right) \left(-\ln\left(\left(1+\mathbb{F}^k_{RN}\right)\right)\right)^\Xi \right)^{\frac{1}{\Xi}}} \right) + \\ \\ \mathring{\mathfrak{i}} \left( -1 + \left( e^{-\left( \sum_{k=1}^2 \left( \frac{1 +^\circ C(\mathfrak{Y}^k_{BF})}{\sum_{k=1}^2 \left(1 +^\circ C(\mathfrak{Y}^k_{BF})\right)} \right) \left(-\ln\left(\left(1+\mathbb{F}^k_{IN}\right)\right)\right)^\Xi \right)^{\frac{1}{\Xi}}} \right) \right) \end{array} \right)$$

Moreover, we consider that the proposed operator is held for $q = Q$, then

$$BCHFAAPG\left(\mathfrak{Y}^1_{BF}, \mathfrak{Y}^2_{BF}, \ldots, \mathfrak{Y}^Q_{BF}\right) = \coprod_{\left( \begin{array}{c} \mathbb{T}^k_{RP}, \mathbb{T}^k_{IP} \in \mathbb{T}_{MS}, \\ \mathbb{F}^k_{RP}, \mathbb{F}^k_{IP} \in \mathbb{F}_{NMS} \end{array} \right)}$$

$$\left( \begin{array}{c} \left( e^{-\left( \sum_{k=1}^Q \left( \frac{1 +^\circ C(\mathfrak{Y}^k_{BF})}{\sum_{k=1}^q \left(1 +^\circ C(\mathfrak{Y}^k_{BF})\right)} \right) \left(-\ln\left(\mathbb{T}^k_{RP}\right)\right)^\Xi \right)^{\frac{1}{\Xi}}} + \mathring{\mathfrak{i}} \left( e^{-\left( \sum_{k=1}^Q \left( \frac{1 +^\circ C(\mathfrak{Y}^k_{BF})}{\sum_{k=1}^q \left(1 +^\circ C(\mathfrak{Y}^k_{BF})\right)} \right) \left(-\ln\left(\mathbb{T}^k_{IP}\right)\right)^\Xi \right)^{\frac{1}{\Xi}}} \right), \\ \\ -1 + \left( e^{-\left( \sum_{k=1}^Q \left( \frac{1 +^\circ C(\mathfrak{Y}^k_{BF})}{\sum_{k=1}^q \left(1 +^\circ C(\mathfrak{Y}^k_{BF})\right)} \right) \left(-\ln\left(\left(1+\mathbb{F}^k_{RN}\right)\right)\right)^\Xi \right)^{\frac{1}{\Xi}}} \right) + \\ \\ \mathring{\mathfrak{i}} \left( -1 + \left( e^{-\left( \sum_{k=1}^Q \left( \frac{1 +^\circ C(\mathfrak{Y}^k_{BF})}{\sum_{k=1}^q \left(1 +^\circ C(\mathfrak{Y}^k_{BF})\right)} \right) \left(-\ln\left(\left(1+\mathbb{F}^k_{IN}\right)\right)\right)^\Xi \right)^{\frac{1}{\Xi}}} \right) \right) \end{array} \right)$$

Thus, we evaluate it for $q = Q + 1$, such as

$$BCHFAAPG\left(\mathfrak{Y}_{BF}^1, \mathfrak{Y}_{BF}^2, \ldots, \mathfrak{Y}_{BF}^{Q+1}\right) = \mathfrak{Y}_{BF}^1 \left(\frac{1 +^\circ C\left(\mathfrak{Y}_{BF}^1\right)}{\sum_{k=1}^{Q+1}\left(1 +^\circ C\left(\mathfrak{Y}_{BF}^k\right)\right)}\right) \otimes \mathfrak{Y}_{BF}^2 \left(\frac{1 +^\circ C\left(\mathfrak{Y}_{BF}^2\right)}{\sum_{k=1}^{Q+1}\left(1 +^\circ C\left(\mathfrak{Y}_{BF}^k\right)\right)}\right)$$

$$\otimes \ldots \otimes \mathfrak{Y}_{BF}^Q \left(\frac{1 +^\circ C\left(\mathfrak{Y}_{BF}^Q\right)}{\sum_{k=1}^{Q+1}\left(1 +^\circ C\left(\mathfrak{Y}_{BF}^k\right)\right)}\right) \otimes \mathfrak{Y}_{BF}^{Q+1} \left(\frac{1 +^\circ C\left(\mathfrak{Y}_{BF}^{Q+1}\right)}{\sum_{k=1}^{Q+1}\left(1 +^\circ C\left(\mathfrak{Y}_{BF}^k\right)\right)}\right)$$

$$= \prod_{k=1}^{Q} \mathfrak{Y}_{BF}^k \left(\frac{1+^\circ C\left(\mathfrak{Y}_{BF}^k\right)}{\sum_{k=1}^{q}\left(1+^\circ C\left(\mathfrak{Y}_{BF}^k\right)\right)}\right) \otimes \mathfrak{Y}_{BF}^{Q+1} \left(\frac{1+^\circ C\left(\mathfrak{Y}_{BF}^{Q+1}\right)}{\sum_{k=1}^{Q+1}\left(1+^\circ C\left(\mathfrak{Y}_{BF}^k\right)\right)}\right)$$

$$= \coprod_{\begin{pmatrix} \mathbb{T}_{RP}^k, \mathbb{T}_{IP}^k \in \mathbb{T}_{MS}, \\ \mathbb{F}_{RP}^k, \mathbb{F}_{IP}^k \in \mathbb{F}_{NMS} \end{pmatrix}}$$

$$\left( \left( e^{-\left(\sum_{k=1}^{Q}\left(\frac{1+^\circ C\left(\mathfrak{Y}_{BF}^k\right)}{\sum_{k=1}^{q}\left(1+^\circ C\left(\mathfrak{Y}_{BF}^k\right)\right)}\right)\left(-\ln\left(\mathbb{T}_{RP}^k\right)\right)^\Xi\right)^{\frac{1}{\Xi}}} + \mathring{\imath}\left(e^{-\left(\sum_{k=1}^{Q}\left(\frac{1+^\circ C\left(\mathfrak{Y}_{BF}^k\right)}{\sum_{k=1}^{q}\left(1+^\circ C\left(\mathfrak{Y}_{BF}^k\right)\right)}\right)\left(-\ln\left(\mathbb{T}_{IP}^k\right)\right)^\Xi\right)^{\frac{1}{\Xi}}}\right), \right.$$

$$-1 + \left(e^{-\left(\sum_{k=1}^{Q}\left(\frac{1+^\circ C\left(\mathfrak{Y}_{BF}^k\right)}{\sum_{k=1}^{q}\left(1+^\circ C\left(\mathfrak{Y}_{BF}^k\right)\right)}\right)\left(-\ln\left(\left(1+\mathbb{F}_{RN}^k\right)\right)\right)^\Xi\right)^{\frac{1}{\Xi}}}\right) +$$

$$\left. \mathring{\imath}\left(-1 + \left(e^{-\left(\sum_{k=1}^{Q}\left(\frac{1+^\circ C\left(\mathfrak{Y}_{BF}^k\right)}{\sum_{k=1}^{q}\left(1+^\circ C\left(\mathfrak{Y}_{BF}^k\right)\right)}\right)\left(-\ln\left(\left(1+\mathbb{F}_{IN}^k\right)\right)\right)^\Xi\right)^{\frac{1}{\Xi}}}\right)\right) \right)$$

$$\otimes \mathfrak{Y}_{BF}^{Q+1} \left(\frac{1 +^\circ C\left(\mathfrak{Y}_{BF}^{Q+1}\right)}{\sum_{k=1}^{Q+1}\left(1 +^\circ C\left(\mathfrak{Y}_{BF}^k\right)\right)}\right)$$

$$= \coprod_{\begin{pmatrix} \mathbb{T}_{RP}^k, \mathbb{T}_{IP}^k \in \mathbb{T}_{MS}, \\ \mathbb{F}_{RP}^k, \mathbb{F}_{IP}^k \in \mathbb{F}_{NMS} \end{pmatrix}}$$

$$\left( \begin{array}{c} \left( e^{-\left( \sum_{k=1}^{Q} \left( \frac{1 +^\circ C(\mathfrak{Y}_{BF}^k)}{\sum_{k=1}^{q} \left(1 +^\circ C(\mathfrak{Y}_{BF}^k)\right)} \right) \left(-\ln\left(\mathbb{T}_{RP}^k\right)\right)^\Xi \right)^{\frac{1}{\Xi}}} + \mathring{\imath} \left( e^{-\left( \sum_{k=1}^{Q} \left( \frac{1 +^\circ C(\mathfrak{Y}_{BF}^k)}{\sum_{k=1}^{q} \left(1 +^\circ C(\mathfrak{Y}_{BF}^k)\right)} \right) \left(-\ln\left(\mathbb{T}_{IP}^k\right)\right)^\Xi \right)^{\frac{1}{\Xi}}} \right), \right) \\ -1 + \left( e^{-\left( \sum_{k=1}^{Q} \left( \frac{1 +^\circ C(\mathfrak{Y}_{BF}^k)}{\sum_{k=1}^{q} \left(1 +^\circ C(\mathfrak{Y}_{BF}^k)\right)} \right) \left(-\ln\left(\left(1 + \mathbb{F}_{RN}^k\right)\right)\right)^\Xi \right)^{\frac{1}{\Xi}}} \right) + \\ \mathring{\imath} \left( -1 + \left( e^{-\left( \sum_{k=1}^{Q} \left( \frac{1 +^\circ C(\mathfrak{Y}_{BF}^k)}{\sum_{k=1}^{q} \left(1 +^\circ C(\mathfrak{Y}_{BF}^k)\right)} \right) \left(-\ln\left(\left(1 + \mathbb{F}_{IN}^k\right)\right)\right)^\Xi \right)^{\frac{1}{\Xi}}} \right) \right) \end{array} \right)$$

$$\otimes \coprod_{\begin{pmatrix} \mathbb{T}_{RP}^{Q+1}, \mathbb{T}_{IP}^{Q+1} \in \mathbb{T}_{MS}, \\ \mathbb{F}_{RP}^{Q+1}, \mathbb{F}_{IP}^{Q+1} \in \mathbb{F}_{NMS} \end{pmatrix}}$$

$$\left( \begin{array}{c} \left( e^{-\left( \left( \frac{1 +^\circ C(\mathfrak{Y}_{BF}^{Q+1})}{\sum_{k=1}^{Q+1} \left(1 +^\circ C(\mathfrak{Y}_{BF}^k)\right)} \right) \left(-\ln\left(\mathbb{T}_{RP}^{Q+1}\right)\right)^\Xi \right)^{\frac{1}{\Xi}}} + \mathring{\imath} \left( e^{-\left( \left( \frac{1 +^\circ C(\mathfrak{Y}_{BF}^{Q+1})}{\sum_{k=1}^{Q+1} \left(1 +^\circ C(\mathfrak{Y}_{BF}^k)\right)} \right) \left(-\ln\left(\mathbb{T}_{IP}^{Q+1}\right)\right)^\Xi \right)^{\frac{1}{\Xi}}} \right), \right) \\ -1 + \left( e^{-\left( \left( \frac{1 +^\circ C(\mathfrak{Y}_{BF}^{Q+1})}{\sum_{k=1}^{Q+1} \left(1 +^\circ C(\mathfrak{Y}_{BF}^k)\right)} \right) \left(-\ln\left(\left(1 + \mathbb{F}_{RN}^{Q+1}\right)\right)\right)^\Xi \right)^{\frac{1}{\Xi}}} \right) + \\ \mathring{\imath} \left( -1 + \left( e^{-\left( \left( \frac{1 +^\circ C(\mathfrak{Y}_{BF}^{Q+1})}{\sum_{k=1}^{Q+1} \left(1 +^\circ C(\mathfrak{Y}_{BF}^k)\right)} \right) \left(-\ln\left(\left(1 + \mathbb{F}_{IN}^{Q+1}\right)\right)\right)^\Xi \right)^{\frac{1}{\Xi}}} \right) \right) \end{array} \right)$$

$$= \coprod_{\begin{pmatrix} \mathbb{T}^k_{RP}, \mathbb{T}^k_{IP} \in \mathbb{T}_{MS}, \\ \mathbb{F}^k_{RP}, \mathbb{F}^k_{IP} \in \mathbb{F}_{NMS} \end{pmatrix}}$$

$$\begin{pmatrix} \left( e^{-\left( \sum_{k=1}^{Q+1} \left( \frac{1 +^\circ C(\mathfrak{Y}^k_{BF})}{\sum_{k=1}^q (1 +^\circ C(\mathfrak{Y}^k_{BF}))} \right) (-\ln(\mathbb{T}^k_{RP}))^\Xi \right)^{\frac{1}{\Xi}}} + \mathring{\mathbb{i}} \left( e^{-\left( \sum_{k=1}^{Q+1} \left( \frac{1 +^\circ C(\mathfrak{Y}^k_{BF})}{\sum_{k=1}^q (1 +^\circ C(\mathfrak{Y}^k_{BF}))} \right) (-\ln(\mathbb{T}^k_{IP}))^\Xi \right)^{\frac{1}{\Xi}}} \right), \\ \\ -1 + \left( e^{-\left( \sum_{k=1}^{Q+1} \left( \frac{1 +^\circ C(\mathfrak{Y}^k_{BF})}{\sum_{k=1}^q (1 +^\circ C(\mathfrak{Y}^k_{BF}))} \right) (-\ln((1+\mathbb{F}^k_{RN})))^\Xi \right)^{\frac{1}{\Xi}}} \right) + \\ \\ \mathring{\mathbb{i}} \left( -1 + \left( e^{-\left( \sum_{k=1}^{Q+1} \left( \frac{1 +^\circ C(\mathfrak{Y}^k_{BF})}{\sum_{k=1}^q (1 +^\circ C(\mathfrak{Y}^k_{BF}))} \right) (-\ln((1+\mathbb{F}^k_{IN})))^\Xi \right)^{\frac{1}{\Xi}}} \right) \right) \end{pmatrix}$$

Hence, our proposed technique is held for all positive integers. Further, we simplify the idempotency, monotonicity, and boundedness of the proposed techniques.

**Proposition 3:** For any collection of BCHFNs. Then

1) When $\mathfrak{Y}^k_{BF} = \mathfrak{Y}_{BF}$, thus

$$BCHFAAPG(\mathfrak{Y}^1_{BF}, \mathfrak{Y}^2_{BF}, \ldots, \mathfrak{Y}^q_{BF}) = \mathfrak{Y}_{BF}.$$

2) When $\mathfrak{Y}^k_{BF} \leq \mathfrak{Y}^{\#k}_{BF}$, thus

$$BCHFAAPG(\mathfrak{Y}^1_{BF}, \mathfrak{Y}^2_{BF}, \ldots, \mathfrak{Y}^q_{BF}) \leq BCHFAAPG\left(\mathfrak{Y}^{\#1}_{BF}, \mathfrak{Y}^{\#2}_{BF}, \ldots, \mathfrak{Y}^{\#q}_{BF}\right).$$

3) When $\mathfrak{Y}^-_{BF} = min(\mathfrak{Y}^1_{BF}, \mathfrak{Y}^2_{BF}, \ldots, \mathfrak{Y}^q_{BF})$ and $\mathfrak{Y}^+_{BF} = max(\mathfrak{Y}^1_{BF}, \mathfrak{Y}^2_{BF}, \ldots, \mathfrak{Y}^q_{BF})$, thus

$$\mathfrak{Y}^-_{BF} \leq BCHFAAPG(\mathfrak{Y}^1_{BF}, \mathfrak{Y}^2_{BF}, \ldots, \mathfrak{Y}^q_{BF}) \leq \mathfrak{Y}^+_{BF}.$$

**Proof:** Omitted.

**Definition 12:** For any collection of BCHFNs, the BCHFAAPWG operator is presented and arranged in the shape:

$$BCHFAAPWG(\mathfrak{Y}^1_{BF}, \mathfrak{Y}^2_{BF}, \ldots, \mathfrak{Y}^q_{BF}) = \mathfrak{Y}^1_{BF} \left( \frac{{}^\circ F^1_{wv}(1 +^\circ C(\mathfrak{Y}^1_{BF}))}{\sum_{k=1}^q {}^\circ F^k_{wv}(1 +^\circ C(\mathfrak{Y}^k_{BF}))} \right) \otimes \mathfrak{Y}^2_{BF} \left( \frac{{}^\circ F^2_{wv}(1 +^\circ C(\mathfrak{Y}^2_{BF}))}{\sum_{k=1}^q {}^\circ F^k_{wv}(1 +^\circ C(\mathfrak{Y}^k_{BF}))} \right)$$

$$\otimes \ldots \otimes \mathfrak{Y}^q_{BF} \left( \frac{{}^\circ F^q_{wv}(1 +^\circ C(\mathfrak{Y}^q_{BF}))}{\sum_{k=1}^q {}^\circ F^k_{wv}(1 +^\circ C(\mathfrak{Y}^k_{BF}))} \right) = \prod_{k=1}^q \mathfrak{Y}^k_{BF} \left( \frac{{}^\circ F^k_{wv}(1 +^\circ C(\mathfrak{Y}^k_{BF}))}{\sum_{k=1}^q {}^\circ F^k_{wv}(1 +^\circ C(\mathfrak{Y}^k_{BF}))} \right)$$

Where ${}^\circ F^k_{wv} \in [0, 1]$, $\sum_{k=1}^q {}^\circ F^k_{wv} = 1$ represents the weight vector.

**Theorem 4:** For any collection of BCHFNs, then we prove that the aggregated value of the above information is again a BCHFN, such as

$$BCHFAAPWG\left(\mathfrak{Y}_{BF}^1, \mathfrak{Y}_{BF}^2, \ldots, \mathfrak{Y}_{BF}^q\right) = \coprod_{\begin{pmatrix} \mathbb{T}_{RP}^k, \mathbb{T}_{IP}^k \in \mathbb{T}_{MS}, \\ \mathbb{F}_{RP}^k, \mathbb{F}_{IP}^k \in \mathbb{F}_{NMS} \end{pmatrix}}$$

$$\left( \begin{array}{c} e^{-\left(\sum_{k=1}^q \left(\frac{{}^{\circ}\mathrm{F}_{wv}^k\left(1 + {}^{\circ} \mathrm{C}\left(\mathfrak{Y}_{BF}^k\right)\right)}{\sum_{k=1}^q {}^{\circ}\mathrm{F}_{wv}^k\left(1 + {}^{\circ} \mathrm{C}\left(\mathfrak{Y}_{BF}^k\right)\right)}\right)\left(-\ln\left(\mathbb{T}_{RP}^k\right)\right)^{\Xi}\right)^{\frac{1}{\Xi}}} + \\[2em]
\mathring{\imath}\left( e^{-\left(\sum_{k=1}^q \left(\frac{{}^{\circ}\mathrm{F}_{wv}^k\left(1 + {}^{\circ} \mathrm{C}\left(\mathfrak{Y}_{BF}^k\right)\right)}{\sum_{k=1}^q {}^{\circ}\mathrm{F}_{wv}^k\left(1 + {}^{\circ} \mathrm{C}\left(\mathfrak{Y}_{BF}^k\right)\right)}\right)\left(-\ln\left(\mathbb{T}_{IP}^k\right)\right)^{\Xi}\right)^{\frac{1}{\Xi}}} \right), \\[2em]
-1 + \left( e^{-\left(\sum_{k=1}^q \left(\frac{{}^{\circ}\mathrm{F}_{wv}^k\left(1 + {}^{\circ} \mathrm{C}\left(\mathfrak{Y}_{BF}^k\right)\right)}{\sum_{k=1}^q {}^{\circ}\mathrm{F}_{wv}^k\left(1 + {}^{\circ} \mathrm{C}\left(\mathfrak{Y}_{BF}^k\right)\right)}\right)\left(-\ln\left(\left(1 + \mathbb{F}_{RN}^k\right)\right)\right)^{\Xi}\right)^{\frac{1}{\Xi}}} \right) + \\[2em]
\mathring{\imath}\left( -1 + \left( e^{-\left(\sum_{k=1}^q \left(\frac{{}^{\circ}\mathrm{F}_{wv}^k\left(1 + {}^{\circ} \mathrm{C}\left(\mathfrak{Y}_{BF}^k\right)\right)}{\sum_{k=1}^q {}^{\circ}\mathrm{F}_{wv}^k\left(1 + {}^{\circ} \mathrm{C}\left(\mathfrak{Y}_{BF}^k\right)\right)}\right)\left(-\ln\left(\left(1 + \mathbb{F}_{IN}^k\right)\right)\right)^{\Xi}\right)^{\frac{1}{\Xi}}} \right) \right) \end{array} \right)$$

Further, we simplify the idempotency, monotonicity, and boundedness of the proposed techniques.

**Proposition 4:** For any collection of BCHFNs. Then

1) When $\mathfrak{Y}_{BF}^k = \mathfrak{Y}_{BF}$, thus

$$BCHFAAPWG\left(\mathfrak{Y}_{BF}^1, \mathfrak{Y}_{BF}^2, \ldots, \mathfrak{Y}_{BF}^q\right) = \mathfrak{Y}_{BF}.$$

2) When $\mathfrak{Y}_{BF}^k \leq \mathfrak{Y}_{BF}^{\#k}$, thus

$$BCHFAAPWG\left(\mathfrak{Y}_{BF}^1, \mathfrak{Y}_{BF}^2, \ldots, \mathfrak{Y}_{BF}^q\right) \leq BCHFAAPWG\left(\mathfrak{Y}_{BF}^{\#1}, \mathfrak{Y}_{BF}^{\#2}, \ldots, \mathfrak{Y}_{BF}^{\#q}\right).$$

3) When $\mathfrak{Y}_{BF}^- = min\left(\mathfrak{Y}_{BF}^1, \mathfrak{Y}_{BF}^2, \ldots, \mathfrak{Y}_{BF}^q\right)$ and $\mathfrak{Y}_{BF}^+ = max\left(\mathfrak{Y}_{BF}^1, \mathfrak{Y}_{BF}^2, \ldots, \mathfrak{Y}_{BF}^q\right)$, thus

$$\mathfrak{Y}_{BF}^- \leq BCHFAAPWG\left(\mathfrak{Y}_{BF}^1, \mathfrak{Y}_{BF}^2, \ldots, \mathfrak{Y}_{BF}^q\right) \leq \mathfrak{Y}_{BF}^+.$$

**Proof:** Omitted.

## 5. MADM procedure for proposed techniques

In this section, we describe the application of the MADM technique based on the BCHFAAPA operator and BCHFAAPG operator to evaluate the supremacy and validity of the invented theory. The geometrical representation of the decision-making technique is listed in Fig 1.

To evaluate the above problem, we consider $\mathfrak{Y}_{BF}^1, \mathfrak{Y}_{BF}^2, \ldots, \mathfrak{Y}_{BF}^q$ as an alternative and for each alternative we have the collection of finite attributes or criteria, such as $\mathfrak{Y}_{AT}^1, \mathfrak{Y}_{AT}^2, \ldots, \mathfrak{Y}_{AT}^p$. Furthermore, to evaluate the above problem, we have an unknown weight vector, because in the proposed operators we have used the power operators instead of the weight vector so because of this reason, we haven't any kind of known weight vector. Additionally, we will assign the value of BCHFN to each attribute in every alternative in a decision matrix, where the term $\mathbb{T}_{MS}(\mathbb{x}) = \left\{ \mathbb{T}_{RP}^j(\mathbb{x}) + \mathring{i}\mathbb{T}_{IP}^j(\mathbb{x}) : j = 1, 2, \ldots, z \right\}$ and $\mathbb{F}_{NMS}(\mathbb{x}) = \left\{ \mathbb{F}_{RP}^j(\mathbb{x}) + \mathring{i}\mathbb{F}_{IP}^j(\mathbb{x}) : j = 1, 2, \ldots, z \right\}$ shows the positive and negative truth grades with a strategy: $\mathbb{T}_{RP}^j, \mathbb{T}_{IP}^j : \mathbb{X} \to [0, 1]$ and $\mathbb{F}_{RP}^j, \mathbb{F}_{IP}^j : \mathbb{X} \to [-1, 0]$. Furthermore, the simple shape of the BCHF number (BCHFN) is stated by: $\mathfrak{Y}_{BF}^k = (\mathbb{T}_{MS}, \mathbb{F}_{NMS}) = \left( \left\{ \mathbb{T}_{RP}^j + \mathring{i}\mathbb{T}_{RP}^j \right\}, \left\{ \mathbb{F}_{RP}^j + \mathring{i}\mathbb{F}_{RP}^j \right\} \right), k = 1, 2, \ldots, q$. Moreover, based on the above information, we demonstrate some practical applications and try to evaluate them with the help of the below decision-making procedure, such as

Step 1: Accumulate the BCHFNs in the form of a decision matrix, if we have cost type of data, then we normalize the matrix, such as

$$N = \begin{cases} \left( \left\{ \mathbb{T}_{RP}^j + \mathring{i}\mathbb{T}_{RP}^j \right\}, \left\{ \mathbb{F}_{RP}^j + \mathring{i}\mathbb{F}_{RP}^j \right\} \right) & \text{for benefit} \\ \left( \left\{ 1 - \mathbb{T}_{RP}^j + \mathring{i}\left( 1 - \mathbb{T}_{RP}^j \right) \right\}, \left\{ -1 - \mathbb{F}_{RP}^j + \mathring{i}\left( -1 - \mathbb{F}_{RP}^j \right) \right\} \right) & \text{for cost} \end{cases}$$

But in the case of benefit type of data, we are not required to be normalized.

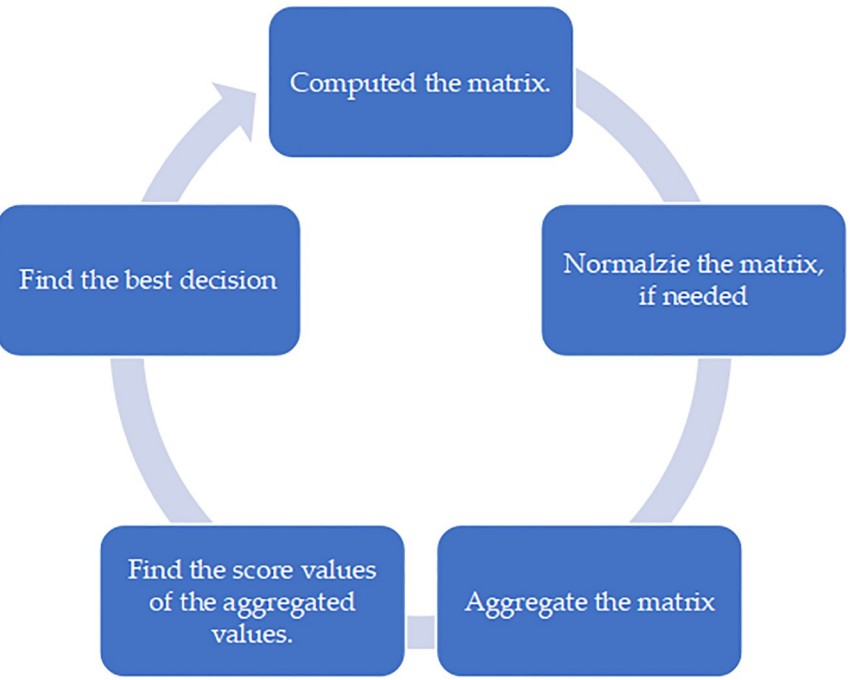

**Fig 1. Graphical interpretation of the proposed algorithm.**

Step 2: After successfully normalization, we aggregate the data by using the BCHFAAPA operator and BCHFAAPG operator, such as

$$
BCHFAAPA\left(\mathfrak{Y}_{BF}^{1}, \mathfrak{Y}_{BF}^{2}, \ldots, \mathfrak{Y}_{BF}^{Q}\right) = \coprod_{\left(\begin{array}{c} \mathbb{T}_{RP}^{k}, \mathbb{T}_{IP}^{k} \in \mathbb{T}_{MS}, \\ \mathbb{F}_{RP}^{k}, \mathbb{F}_{IP}^{k} \in \mathbb{F}_{NMS} \end{array}\right)}
$$

$$
\left(\begin{array}{c}
\left(1 - e^{-\left(\sum_{k=1}^{Q}\left(\frac{1 +^{\circ} C\left(\mathfrak{Y}_{BF}^{k}\right)}{\sum_{k=1}^{q}\left(1 +^{\circ} C\left(\mathfrak{Y}_{BF}^{k}\right)\right)}\right)\left(-\ln\left(1 - \mathbb{T}_{RP}^{k}\right)\right)^{\Xi}\right)^{\frac{1}{\Xi}}} + \mathring{\mathbb{i}}\left(1 - e^{-\left(\sum_{k=1}^{Q}\left(\frac{1 +^{\circ} C\left(\mathfrak{Y}_{BF}^{k}\right)}{\sum_{k=1}^{q}\left(1 +^{\circ} C\left(\mathfrak{Y}_{BF}^{k}\right)\right)}\right)\left(-\ln\left(1 - \mathbb{T}_{IP}^{k}\right)\right)^{\Xi}\right)^{\frac{1}{\Xi}}}\right)\right), \\
\left(-\left(e^{-\left(\sum_{k=1}^{Q}\left(\frac{1 +^{\circ} C\left(\mathfrak{Y}_{BF}^{k}\right)}{\sum_{k=1}^{q}\left(1 +^{\circ} C\left(\mathfrak{Y}_{BF}^{k}\right)\right)}\right)\left(-\ln\left(\left|\mathbb{F}_{RN}^{k}\right|\right)\right)^{\Xi}\right)^{\frac{1}{\Xi}}}\right) + \mathring{\mathbb{i}}\left(-\left(e^{-\left(\sum_{k=1}^{Q}\left(\frac{1 +^{\circ} C\left(\mathfrak{Y}_{BF}^{k}\right)}{\sum_{k=1}^{q}\left(1 +^{\circ} C\left(\mathfrak{Y}_{BF}^{k}\right)\right)}\right)\left(-\ln\left(\left|\mathbb{F}_{IN}^{k}\right|\right)\right)^{\Xi}\right)^{\frac{1}{\Xi}}}\right)\right)\right)
\end{array}\right)
$$

$$
BCHFAAPG\left(\mathfrak{Y}_{BF}^{1}, \mathfrak{Y}_{BF}^{2}, \ldots, \mathfrak{Y}_{BF}^{q}\right) = \coprod_{\left(\begin{array}{c} \mathbb{T}_{RP}^{k}, \mathbb{T}_{IP}^{k} \in \mathbb{T}_{MS}, \\ \mathbb{F}_{RP}^{k}, \mathbb{F}_{IP}^{k} \in \mathbb{F}_{NMS} \end{array}\right)}
$$

$$
\left(\begin{array}{c}
\left(e^{-\left(\sum_{k=1}^{q}\left(\frac{1 +^{\circ} C\left(\mathfrak{Y}_{BF}^{k}\right)}{\sum_{k=1}^{q}\left(1 +^{\circ} C\left(\mathfrak{Y}_{BF}^{k}\right)\right)}\right)\left(-\ln\left(\mathbb{T}_{RP}^{k}\right)\right)^{\Xi}\right)^{\frac{1}{\Xi}}} + \mathring{\mathbb{i}}\left(e^{-\left(\sum_{k=1}^{q}\left(\frac{1 +^{\circ} C\left(\mathfrak{Y}_{BF}^{k}\right)}{\sum_{k=1}^{q}\left(1 +^{\circ} C\left(\mathfrak{Y}_{BF}^{k}\right)\right)}\right)\left(-\ln\left(\mathbb{T}_{IP}^{k}\right)\right)^{\Xi}\right)^{\frac{1}{\Xi}}}\right)\right), \\
-1 + \left(e^{-\left(\sum_{k=1}^{q}\left(\frac{1 +^{\circ} C\left(\mathfrak{Y}_{BF}^{k}\right)}{\sum_{k=1}^{q}\left(1 +^{\circ} C\left(\mathfrak{Y}_{BF}^{k}\right)\right)}\right)\left(-\ln\left(\left(1 + \mathbb{F}_{RN}^{k}\right)\right)\right)^{\Xi}\right)^{\frac{1}{\Xi}}}\right) + \\
\mathring{\mathbb{i}}\left(-1 + \left(e^{-\left(\sum_{k=1}^{q}\left(\frac{1 +^{\circ} C\left(\mathfrak{Y}_{BF}^{k}\right)}{\sum_{k=1}^{q}\left(1 +^{\circ} C\left(\mathfrak{Y}_{BF}^{k}\right)\right)}\right)\left(-\ln\left(\left(1 + \mathbb{F}_{IN}^{k}\right)\right)\right)^{\Xi}\right)^{\frac{1}{\Xi}}}\right)\right)
\end{array}\right)
$$

Step 3: After evaluating the aggregation information, we find the score values, such as

$$
S\left(\mathfrak{Y}_{BF}^{1}\right) = \frac{1}{2}\left(\begin{array}{c} \frac{1}{order\ of\ \left(\mathbb{T}_{RP}^{j}\right)}\sum_{j=1}^{z}\mathbb{T}_{RP}^{j} + \frac{1}{order\ of\ \left(\mathbb{T}_{IP}^{j}\right)}\sum_{j=1}^{z}\mathbb{T}_{IP}^{j} - \\ \frac{1}{order\ of\ \left(\mathbb{F}_{RP}^{j}\right)}\sum_{j=1}^{z}\mathbb{F}_{RP}^{j} - \frac{1}{order\ of\ \left(\mathbb{F}_{IP}^{j}\right)}\sum_{j=1}^{z}\mathbb{F}_{IP}^{j} \end{array}\right) \in [-1, 1]
$$

If the idea of score values has been failed, then we will use the accuracy values, such as

$$H\left(\mathfrak{Y}_{BF}^{1}\right) = \frac{1}{2}\left( \begin{array}{c} \dfrac{1}{order\ of\ \left(\mathbb{T}_{RP}^{j}\right)}\sum_{j=1}^{z}\mathbb{T}_{RP}^{j} + \dfrac{1}{order\ of\ \left(\mathbb{T}_{IP}^{j}\right)}\sum_{j=1}^{z}\mathbb{T}_{IP}^{j} + \\ \dfrac{1}{order\ of\ \left(\mathbb{F}_{RP}^{j}\right)}\sum_{j=1}^{z}\mathbb{F}_{RP}^{j} + \dfrac{1}{order\ of\ \left(\mathbb{F}_{IP}^{j}\right)}\sum_{j=1}^{z}\mathbb{F}_{IP}^{j} \end{array} \right) \in [0, 1]$$

Step 4: Finally, to find the best optimal, we rank all the alternatives based on the score values.

For implementing the above procedure, we have been required to discuss some real-life problems, for this, we consider the problems of LMS in higher education implementation based on the proposed work especially in China, because during COVID-19 they faced a lot of problems, therefore, based on the proposed techniques we discussed the application of LMS in higher educations.

## 5.1. Implementation of LMS in higher education systems through proposed operators: A case review for China

In this sub-section, we aim to discuss the different features or factors of the LMS in higher education systems based on initiated techniques. For this, we consider the problem of LMS in China's education systems. Therefore, by using the initiated technique, we aim to evaluate the best or most preferable factor that plays an important role in the implementation of the LMS in education or non-education departments in China. The major impact of this application is to review the utilization of the LMSs in higher education in Chains, for this, we have the following five alternatives, such as

1. Improved Accessibility "$\mathfrak{Y}_{BF}^{1}$"

2. Blended Learning "$\mathfrak{Y}_{BF}^{2}$"

3. Collaboration and Communications "$\mathfrak{Y}_{BF}^{3}$"

4. Assessment and Evaluation "$\mathfrak{Y}_{BF}^{4}$"

5. Administrative Efficiency "$\mathfrak{Y}_{BF}^{5}$"

Overall, the implementation of the LMS in higher education in China is a more valuable and dominant step towards success, because, with the help of this utilization, we improve the quality and accessibility of education in the country. Based on the proposed operators we find the most perfect and most valuable feature among the above five, we use the following features, such as growth analysis, social impact, political impact, environmental impact, and education ratio of the people in each region. Therefore, based on the above information, we demonstrate some practical applications and try to evaluate them with the help of the below decision-making procedure, such as

Step 1: Accumulate the BCHFNs in the form of a decision matrix, if we have cost type of data, then we normalize the matrix, such as

$$N = \begin{cases} \left(\left\{\mathbb{T}_{RP}^{j} + \mathring{\mathbb{i}}\mathbb{T}_{RP}^{j}\right\}, \left\{\mathbb{F}_{RP}^{j} + \mathring{\mathbb{i}}\mathbb{F}_{RP}^{j}\right\}\right) & \textit{for benefit} \\ \left(\left\{1 - \mathbb{T}_{RP}^{j} + \mathring{\mathbb{i}}\left(1 - \mathbb{T}_{RP}^{j}\right)\right\}, \left\{-1 - \mathbb{F}_{RP}^{j} + \mathring{\mathbb{i}}\left(-1 - \mathbb{F}_{RP}^{j}\right)\right\}\right) & \textit{for cost} \end{cases}$$

But in the case of benefit type of data, we are not required to be normalized. However, the selected information is not required to be normalized.

$$
\mathfrak{Y}_{BF}^1 = \left\{
\begin{array}{l}
\left(\mathfrak{Y}_{AT}^1, \{0.8 + \mathring{\imath}0.9, 0.7 + \mathring{\imath}0.8, 0.6 + \mathring{\imath}0.8\}, \{-0.4 + \mathring{\imath}(-0.5), -0.5 + \mathring{\imath}(-0.6), -0.6 + \mathring{\imath}(-0.7)\}\right), \\
\left(\mathfrak{Y}_{AT}^2, \{0.81 + \mathring{\imath}0.91, 0.71 + \mathring{\imath}0.81, 0.61 + \mathring{\imath}0.81\}, \{-0.41 + \mathring{\imath}(-0.51), -0.51 + \mathring{\imath}(-0.61), -0.61 + \mathring{\imath}(-0.71)\}\right), \\
\left(\mathfrak{Y}_{AT}^3, \{0.82 + \mathring{\imath}0.92, 0.72 + \mathring{\imath}0.82, 0.62 + \mathring{\imath}0.82\}, \{-0.42 + \mathring{\imath}(-0.52), -0.52 + \mathring{\imath}(-0.62), -0.62 + \mathring{\imath}(-0.72)\}\right), \\
\left(\mathfrak{Y}_{AT}^4, \{0.83 + \mathring{\imath}0.93, 0.73 + \mathring{\imath}0.83, 0.63 + \mathring{\imath}0.83\}, \{-0.43 + \mathring{\imath}(-0.53), -0.53 + \mathring{\imath}(-0.63), -0.63 + \mathring{\imath}(-0.73)\}\right), \\
\left(\mathfrak{Y}_{AT}^5, \{0.84 + \mathring{\imath}0.94, 0.74 + \mathring{\imath}0.84, 0.64 + \mathring{\imath}0.84\}, \{-0.44 + \mathring{\imath}(-0.54), -0.54 + \mathring{\imath}(-0.64), -0.64 + \mathring{\imath}(-0.74)\}\right)
\end{array}
\right\}
$$

$$
\mathfrak{Y}_{BF}^2 = \left\{
\begin{array}{l}
\left(\mathfrak{Y}_{AT}^1, \{0.4 + \mathring{\imath}0.5, 0.3 + \mathring{\imath}0.4, 0.6 + \mathring{\imath}0.1\}, \{-0.2 + \mathring{\imath}(-0.3), -0.4 + \mathring{\imath}(-0.2), -0.5 + \mathring{\imath}(-0.2)\}\right), \\
\left(\mathfrak{Y}_{AT}^2, \{0.41 + \mathring{\imath}0.51, 0.31 + \mathring{\imath}0.41, 0.61 + \mathring{\imath}0.11\}, \{-0.21 + \mathring{\imath}(-0.31), -0.41 + \mathring{\imath}(-0.21), -0.51 + \mathring{\imath}(-0.21)\}\right), \\
\left(\mathfrak{Y}_{AT}^3, \{0.42 + \mathring{\imath}0.52, 0.32 + \mathring{\imath}0.42, 0.62 + \mathring{\imath}0.12\}, \{-0.22 + \mathring{\imath}(-0.32), -0.42 + \mathring{\imath}(-0.22), -0.52 + \mathring{\imath}(-0.22)\}\right), \\
\left(\mathfrak{Y}_{AT}^4, \{0.43 + \mathring{\imath}0.53, 0.33 + \mathring{\imath}0.43, 0.63 + \mathring{\imath}0.13\}, \{-0.23 + \mathring{\imath}(-0.33), -0.43 + \mathring{\imath}(-0.23), -0.53 + \mathring{\imath}(-0.23)\}\right), \\
\left(\mathfrak{Y}_{AT}^5, \{0.44 + \mathring{\imath}0.54, 0.34 + \mathring{\imath}0.44, 0.64 + \mathring{\imath}0.14\}, \{-0.24 + \mathring{\imath}(-0.34), -0.44 + \mathring{\imath}(-0.24), -0.54 + \mathring{\imath}(-0.24)\}\right)
\end{array}
\right\}
$$

$$
\mathfrak{Y}_{BF}^3 = \left\{
\begin{array}{l}
\left(\mathfrak{Y}_{AT}^1, \{0.1 + \mathring{\imath}0.2, 0.3 + \mathring{\imath}0.4, 0.5 + \mathring{\imath}0.6\}, \{-0.7 + \mathring{\imath}(-0.6), -0.4 + \mathring{\imath}(-0.3), -0.2 + \mathring{\imath}(-0.2)\}\right), \\
\left(\mathfrak{Y}_{AT}^2, \{0.11 + \mathring{\imath}0.21, 0.31 + \mathring{\imath}0.41, 0.51 + \mathring{\imath}0.61\}, \{-0.71 + \mathring{\imath}(-0.61), -0.41 + \mathring{\imath}(-0.31), -0.21 + \mathring{\imath}(-0.21)\}\right), \\
\left(\mathfrak{Y}_{AT}^3, \{0.12 + \mathring{\imath}0.22, 0.32 + \mathring{\imath}0.42, 0.52 + \mathring{\imath}0.62\}, \{-0.72 + \mathring{\imath}(-0.62), -0.42 + \mathring{\imath}(-0.32), -0.22 + \mathring{\imath}(-0.22)\}\right), \\
\left(\mathfrak{Y}_{AT}^4, \{0.13 + \mathring{\imath}0.23, 0.33 + \mathring{\imath}0.43, 0.53 + \mathring{\imath}0.63\}, \{-0.73 + \mathring{\imath}(-0.63), -0.43 + \mathring{\imath}(-0.33), -0.23 + \mathring{\imath}(-0.23)\}\right), \\
\left(\mathfrak{Y}_{AT}^5, \{0.14 + \mathring{\imath}0.24, 0.34 + \mathring{\imath}0.44, 0.54 + \mathring{\imath}0.64\}, \{-0.74 + \mathring{\imath}(-0.64), -0.44 + \mathring{\imath}(-0.34), -0.24 + \mathring{\imath}(-0.24)\}\right)
\end{array}
\right\}
$$

$$
\mathfrak{Y}_{BF}^4 = \left\{
\begin{array}{l}
\left(\mathfrak{Y}_{AT}^1, \{0.4 + \mathring{\imath}0.5, 0.6 + \mathring{\imath}0.7, 0.8 + \mathring{\imath}0.9\}, \{-0.4 + \mathring{\imath}(-0.5), -0.3 + \mathring{\imath}(-0.2), -0.1 + \mathring{\imath}(-0.2)\}\right), \\
\left(\mathfrak{Y}_{AT}^2, \{0.41 + \mathring{\imath}0.51, 0.61 + \mathring{\imath}0.71, 0.81 + \mathring{\imath}0.91\}, \{-0.41 + \mathring{\imath}(-0.51), -0.31 + \mathring{\imath}(-0.21), -0.11 + \mathring{\imath}(-0.21)\}\right), \\
\left(\mathfrak{Y}_{AT}^3, \{0.42 + \mathring{\imath}0.52, 0.62 + \mathring{\imath}0.72, 0.82 + \mathring{\imath}0.92\}, \{-0.42 + \mathring{\imath}(-0.52), -0.32 + \mathring{\imath}(-0.22), -0.122 + \mathring{\imath}(-0.22)\}\right), \\
\left(\mathfrak{Y}_{AT}^4, \{0.43 + \mathring{\imath}0.53, 0.63 + \mathring{\imath}0.73, 0.83 + \mathring{\imath}0.93\}, \{-0.43 + \mathring{\imath}(-0.53), -0.33 + \mathring{\imath}(-0.23), -0.13 + \mathring{\imath}(-0.23)\}\right), \\
\left(\mathfrak{Y}_{AT}^5, \{0.44 + \mathring{\imath}0.54, 0.64 + \mathring{\imath}0.74, 0.84 + \mathring{\imath}0.94\}, \{-0.44 + \mathring{\imath}(-0.54), -0.34 + \mathring{\imath}(-0.24), -0.14 + \mathring{\imath}(-0.24)\}\right)
\end{array}
\right\}
$$

$$
\mathfrak{Y}_{BF}^5 = \left\{
\begin{array}{l}
\left(\mathfrak{Y}_{AT}^1, \{0.3 + \mathring{\imath}0.4, 0.5 + \mathring{\imath}0.2, 0.3 + \mathring{\imath}0.4\}, \{-0.1 + \mathring{\imath}(-0.2), -0.3 + \mathring{\imath}(-0.4), -0.5 + \mathring{\imath}(-0.4)\}\right), \\
\left(\mathfrak{Y}_{AT}^2, \{0.31 + \mathring{\imath}0.41, 0.51 + \mathring{\imath}0.21, 0.31 + \mathring{\imath}0.41\}, \{-0.11 + \mathring{\imath}(-0.21), -0.31 + \mathring{\imath}(-0.41), -0.51 + \mathring{\imath}(-0.41)\}\right), \\
\left(\mathfrak{Y}_{AT}^3, \{0.32 + \mathring{\imath}0.42, 0.52 + \mathring{\imath}0.22, 0.32 + \mathring{\imath}0.42\}, \{-0.12 + \mathring{\imath}(-0.22), -0.32 + \mathring{\imath}(-0.42), -0.52 + \mathring{\imath}(-0.42)\}\right), \\
\left(\mathfrak{Y}_{AT}^4, \{0.33 + \mathring{\imath}0.43, 0.53 + \mathring{\imath}0.23, 0.33 + \mathring{\imath}0.43\}, \{-0.13 + \mathring{\imath}(-0.23), -0.33 + \mathring{\imath}(-0.43), -0.53 + \mathring{\imath}(-0.43)\}\right), \\
\left(\mathfrak{Y}_{AT}^5, \{0.34 + \mathring{\imath}0.44, 0.54 + \mathring{\imath}0.24, 0.34 + \mathring{\imath}0.44\}, \{-0.14 + \mathring{\imath}(-0.24), -0.34 + \mathring{\imath}(-0.44), -0.54 + \mathring{\imath}(-0.44)\}\right)
\end{array}
\right\}
$$

Step 2: After successfully normalization, we aggregate the data by using the BCHFAAPA

operator, such as

$$\mathfrak{Y}_{BF}^1 = \left\{ \left( \begin{array}{c} \{0.8209 + \hat{\imath}0.9218, 0.7206 + \hat{\imath}0.8209, 0.6205 + \hat{\imath}0.8209\}, \\ \{-0.4194 + \hat{\imath}(-0.5195), -0.5195 + \hat{\imath}(-0.6194), -0.6194 + \hat{\imath}(-0.7194)\} \end{array} \right) \right\}$$

$$\mathfrak{Y}_{BF}^2 = \left\{ \left( \begin{array}{c} \{0.4205 + \hat{\imath}0.5205, 0.3205 + \hat{\imath}0.4205, 0.6205 + \hat{\imath}0.1210\}, \\ \{-0.2192 + \hat{\imath}(-0.3194), -0.4194 + \hat{\imath}(-0.2192), -0.5195 + \hat{\imath}(-0.2192)\} \end{array} \right) \right\}$$

$$\mathfrak{Y}_{BF}^3 = \left\{ \left( \begin{array}{c} \{0.1210 + \hat{\imath}0.2206, 0.3205 + \hat{\imath}0.4205, 0.5205 + \hat{\imath}0.6205\}, \\ \{-0.7194 + \hat{\imath}(-0.6194), -0.4194 + \hat{\imath}(-0.3194), -0.2192 + \hat{\imath}(-0.2192)\} \end{array} \right) \right\}$$

$$\mathfrak{Y}_{BF}^4 = \left\{ \left( \begin{array}{c} \{0.4205 + \hat{\imath}0.5205, 0.6205 + \hat{\imath}0.7206, 0.8209 + \hat{\imath}0.9218\}, \\ \{-0.4194 + \hat{\imath}(-0.5195), -0.3194 + \hat{\imath}(-0.2192), -0.1187 + \hat{\imath}(-0.2192)\} \end{array} \right) \right\}$$

$$\mathfrak{Y}_{BF}^5 = \left\{ \left( \begin{array}{c} \{0.3205 + \hat{\imath}0.4205, 0.5205 + \hat{\imath}0.2206, 0.3205 + \hat{\imath}0.4205\}, \\ \{-0.1187 + \hat{\imath}(-0.2192), -0.3194 + \hat{\imath}(-0.4194), -0.5195 + \hat{\imath}(-0.4194)\} \end{array} \right) \right\}$$

Similarly, we aggregate the data by using the BCHFAAPG operator, such as

$$\mathfrak{Y}_{BF}^1 = \left\{ \left( \begin{array}{c} \{0.8192 + \hat{\imath}0.9185, 0.7194 + \hat{\imath}0.8192, 0.6194 + \hat{\imath}0.8192\}, \\ \{-0.4205 + \hat{\imath}(-0.5205), -0.5205 + \hat{\imath}(-0.62055), -0.6205 + \hat{\imath}(-0.7206)\} \end{array} \right) \right\}$$

$$\mathfrak{Y}_{BF}^2 = \left\{ \left( \begin{array}{c} \{0.4194 + \hat{\imath}0.5195, 0.3194 + \hat{\imath}0.4194, 0.6194 + \hat{\imath}0.1187\}, \\ \{-0.2206 + \hat{\imath}(-0.3205), -0.4205 + \hat{\imath}(-0.2206), -0.5205 + \hat{\imath}(-0.2206)\} \end{array} \right) \right\}$$

$$\mathfrak{Y}_{BF}^3 = \left\{ \left( \begin{array}{c} \{0.1187 + \hat{\imath}0.2192, 0.3194 + \hat{\imath}0.4194, 0.5195 + \hat{\imath}0.6194\}, \\ \{-0.7206 + \hat{\imath}(-0.6205), -0.4205 + \hat{\imath}(-0.3205), -0.2206 + \hat{\imath}(-0.2206)\} \end{array} \right) \right\}$$

$$\mathfrak{Y}_{BF}^4 = \left\{ \left( \begin{array}{c} \{0.4194 + \hat{\imath}0.5195, 0.6194 + \hat{\imath}0.7194, 0.8192 + \hat{\imath}0.9185\}, \\ \{-0.4205 + \hat{\imath}(-0.5205), -0.3205 + \hat{\imath}(-0.2206), -0.1210 + \hat{\imath}(-0.2206)\} \end{array} \right) \right\}$$

$$\mathfrak{Y}_{BF}^5 = \left\{ \left( \begin{array}{c} \{0.3194 + \hat{\imath}0.4194, 0.5195 + \hat{\imath}0.2192, 0.3194 + \hat{\imath}0.4194\}, \\ \{-0.1210 + \hat{\imath}(-0.2206), -0.3205 + \hat{\imath}(-0.4205), -0.5205 + \hat{\imath}(-0.4205)\} \end{array} \right) \right\}$$

Step 3: After evaluating the aggregation information, we find the score values, such as
For BCHFAAPA operators, such as

$$\mathfrak{Y}_{BF}^1 = 0.6785, \mathfrak{Y}_{BF}^2 = 0.36164, \mathfrak{Y}_{BF}^3 = 0.395, \mathfrak{Y}_{BF}^4 = 0.4867, \mathfrak{Y}_{BF}^5 = 0.3532$$

For BCHFAAPG operators, such as

$$\mathfrak{Y}_{BF}^1 = 0.6782, \mathfrak{Y}_{BF}^2 = 0.36163, \mathfrak{Y}_{BF}^3 = 0.3949, \mathfrak{Y}_{BF}^4 = 0.4866, \mathfrak{Y}_{BF}^5 = 0.3533$$

Step 4: Finally, to find the best optimal, we rank all the alternatives based on the score values, such as

For BCHFAAPA operators, such as

$$\mathfrak{Y}_{BF}^1 > \mathfrak{Y}_{BF}^4 > \mathfrak{Y}_{BF}^3 > \mathfrak{Y}_{BF}^2 > \mathfrak{Y}_{BF}^5$$

For BCHFAAPG operators, such as

$$\mathfrak{Y}_{BF}^1 > \mathfrak{Y}_{BF}^4 > \mathfrak{Y}_{BF}^3 > \mathfrak{Y}_{BF}^2 > \mathfrak{Y}_{BF}^5$$

The best and most valuable decision is $\mathfrak{Y}_{BF}^1$ according to the theory of the BCHFAAPA operator and BCHFAAPG operator, which represents improved accessibility, where the implementation of the LMS in higher education in China is dependent on improved accessibility. Furthermore, we verify the supremacy and validity of the proposed techniques by comparing our results with some existing techniques.

## 6. Comparative analysis

In this section, under the presence of the data in the above section, we compare the proposed ranking results with the obtained ranking results of the prevailing techniques. For this, we are making a comparison between the proposed techniques with some existing operators to enhance the worth of the proposed operators. Therefore, we noticed that no one could derive the idea of power aggregation operators [25] based on Aczel-Alsina operational laws. For comparison, we have the following prevailing operators, such as Senapati et al. [26] derived the Aczel-Alsina operators for HFSs, Mahmood et al. [27] evaluated the Aczel-Alsina operators for BCFSs, Garg et al. [28] invented the Aczel-Alsina power operators for BFSs, Mahmood et al. [29] examined the geometric Aczel-Alsina operators for BCFSs, Hayat et al. [30] exposed the aggregation operators for q-rung orthopair fuzzy sets, and Yang et al. [31] invented the interaction operators for q-rung orthopair fuzzy soft sets. The comparative analysis between proposed and prevailing operators is stated in Table 1.

The best and most valuable decision is $\mathfrak{Y}_{BF}^1$ according to the theory of the BCHFAAPA operator and BCHFAAPG operator, which represents improved accessibility, where the implementation of the LMS in higher education in China is dependent on improved accessibility. But the existing techniques in [25–31] are not working accurately and correctly, because our proposed operators are computed based on BCHFSs which are fully novel and new, yet no one could define any kind of operators based on theme, therefore, these all-existing techniques are the specials cases of the proposed operators based on BCHFSs.

Table 1. Comparison between proposed and existing methods.

| Methods | Score values | Ranking values |
|---|---|---|
| Yager [25] | Not able to evaluate the proposed data | Failed |
| Senapati et al. [26] | Not able to evaluate the proposed data | Failed |
| Mahmood et al. [27] | Not able to evaluate the proposed data | Failed |
| Garg et al. [28] | Not able to evaluate the proposed data | Failed |
| Mahmood et al. [29] | Not able to evaluate the proposed data | Failed |
| Hayat et al. [30] | Not able to evaluate the proposed data | Failed |
| Yang et al. [31] | Not able to evaluate the proposed data | Failed |
| BCHFAAA operator | $\mathfrak{Y}_{BF}^1 = 0.6785, \mathfrak{Y}_{BF}^2 = 0.36164,$ $\mathfrak{Y}_{BF}^3 = 0.395, \mathfrak{Y}_{BF}^4 = 0.4867, \mathfrak{Y}_{BF}^5 = 0.3532$ | $\mathfrak{Y}_{BF}^1 > \mathfrak{Y}_{BF}^4 > \mathfrak{Y}_{BF}^3 > \mathfrak{Y}_{BF}^2 > \mathfrak{Y}_{BF}^5$ |
| BCHFAAPG operator | $\mathfrak{Y}_{BF}^1 = 0.6782, \mathfrak{Y}_{BF}^2 = 0.36163,$ $\mathfrak{Y}_{BF}^3 = 0.3949, \mathfrak{Y}_{BF}^4 = 0.4866, \mathfrak{Y}_{BF}^5 = 0.3533$ | $\mathfrak{Y}_{BF}^1 > \mathfrak{Y}_{BF}^4 > \mathfrak{Y}_{BF}^3 > \mathfrak{Y}_{BF}^2 > \mathfrak{Y}_{BF}^5$ |

## 7. Conclusion

The major contribution of the proposed theory is listed below:

1. We derived the techniques of BCHF sets, then we evaluated some flexible operational laws, called Algebraic operational laws and Aczel-Alsina operational laws.

2. We elaborated on the techniques of BCHFAAPA, BCHFAAPWA, BCHFAAPG, and BCHFAAPWG operators.

3. We discussed some basic properties of each proposed operator.

4. We computed the MADM techniques for invented operators.

5. We selected some prevailing operators and tried to compare their ranking results with our proposed results to enhance the worth and capability of the invented theory.

In the future, we will expose some new ideas such as prioritized Aczel-Alsina operators [32] based on BCHFS and their extensions [33, 34]. Further, we will improve their worth with the help of the utilization of the proposed theory in the field of artificial intelligence [35], machine learning [36], education department [37], and many others to enhance the worth of the derived theory.

## Author Contributions

**Methodology:** Shi Yin.

**Project administration:** Shi Yin.

**Resources:** Zeeshan Ali.

**Software:** Zeeshan Ali.

**Writing – original draft:** Lijun Ma, Shi Yin.

**Writing – review & editing:** Lijun Ma, Zeeshan Ali, Shi Yin.

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
