## [Decision Letter · Decision Letter 0]

25 Jan 2024

PONE-D-23-39846

Implementation of Learning Management Systems (LMS) in Higher Education Systems Through Bipolar Complex Hesitant Fuzzy Aczel-Alsina Power Aggregation Operators: A Case Review for China

PLOS ONE

Dear Dr. Yin,

Thank you for submitting your manuscript to PLOS ONE. After careful consideration, we feel that it has merit but does not fully meet PLOS ONE’s publication criteria as it currently stands. Therefore, we invite you to submit a revised version of the manuscript that addresses the points raised during the review process.

We look forward to receiving your revised manuscript.

Kind regards,

Tien V.T. Nguyen

Academic Editor

PLOS ONE

Journal Requirements:

3. In the online submission form, you indicated that "The data presented in this study are available on request from the corresponding author."

Additional Editor Comments:

Dear Prof. Shi Yin:

Thank you for your submission.

The manuscript explores the application of bipolar complex hesitant fuzzy (BCHF) sets and associated operational laws, particularly the Aczel-Alsina operational laws, in evaluating the significance of Learning Management Systems (LMSs) in diverse fields. The authors introduce novel operators like BCHF Aczel-Alsina power averaging (BCHFAAPA) and BCHF Aczel-Alsina power weighted averaging (BCHFAAPWA) to assess the importance of LMSs. Additionally, multi-attribute decision-making (MADM) techniques are employed to evaluate the key factors influencing LMS utilization.

-We have carefully evaluated your work, and we are impressed with the clarity of organization, the quality of writing, and the substantial contributions your manuscript makes to the field. Your innovative approach in applying bipolar complex hesitant fuzzy sets and associated operators to evaluate the importance of Learning Management Systems showcases a significant advancement in the understanding of decision-making processes within various sectors.

-We believe that your research will make a valuable addition to academic literature, providing insights and methodologies that will benefit scholars, researchers, and practitioners alike.

-The acceptance of your manuscript is contingent upon addressing any minor editorial suggestions or formatting adjustments that the editorial team may communicate to you shortly. Once these revisions are made, we look forward to proceeding with the publication process.

-We would like to express our appreciation for your hard work and dedication to producing a high-quality manuscript. Congratulations on this achievement, and we anticipate the positive impact your work will have on the academic community.

Thank you for choosing Plos ONE as the platform to disseminate your research.

Best regards,

Tien V.T. Nguyen, Ph.D.

Academic Editor

Reviewers' comments:

Reviewer's Responses to Questions

**Comments to the Author**

1. Is the manuscript technically sound, and do the data support the conclusions?

Reviewer #1: Yes

Reviewer #2: Yes

2. Has the statistical analysis been performed appropriately and rigorously? 

Reviewer #1: Yes

Reviewer #2: Yes

3. Have the authors made all data underlying the findings in their manuscript fully available?

Reviewer #1: Yes

Reviewer #2: Yes

4. Is the manuscript presented in an intelligible fashion and written in standard English?

Reviewer #1: Yes

Reviewer #2: Yes

5. Review Comments to the Author

Reviewer #1: Please read my comments/suggestions given below for preparing the revised draft:

My Comments and Suggestions to Authors:

1-The abstract is not convincing and is disorganized, it should be refined to precisely illustrate what authors have done in this paper within 200 words. The abstract must be a concise yet comprehensive reflection of what is in your paper. Remember that reader want to know: 1-what is the problem. 2- why the problem is relevant 3- wants an overview of your approach. 4-need to know the results.

1-Manuscript needs a good introduction, the introduction section of the manuscript is weak, authors are advised to improvise the introduction section.

2-In the Introduction part, the new features of the proposed method and the main advantages of the results over others should be clearly described.

3-An introduction should clearly highlight the motivation, problem statement, the objective of the paper, gap in the existing research and the novelty of the conducted research.

4-This application topic has not received much attention in the literature. However, the study, literature review and presentation require substantial improvement in several respects.

5-How are the parameters in the proposed model selected?

6-The problems in this manuscript are not clear.

7-There are no citations for many sentences in this manuscript. Why? Please check.

8-Some variables weren't defined appropriately.

9-English language needs to be improved significantly.

10-Paper organization needs to be restructured. Kindly be as concise and straight to the point as possible.

11-The writing of the manuscript. There are a many incomplete sentences or sentences without subjects.

12-Many details are missing and others unclear.

13-The database requires more explanation.

14-Result and Discussion section is inadequate. Need more attention and better explanation.

15-When I checked the results, I noticed that there were mistakes, please recheck

16-The results are not easy to follow.

17-I suggest extending the conclusions section to focus on the results you get, the method you propose, and their significance.

18-There are many repetitions in sentences.

19-References aren't formatted according to rules

20-Some articles listed in References are old, we are in 2024.

Reviewer #2: In this paper author addressed Implementation of Learning Management Systems (LMS) in Higher Education Systems

Through Bipolar Complex Hesitant Fuzzy Aczel-Alsina Power Aggregation Operators: A Case Review for China. The paper needs following revision;

1. Comparison of the paper should be extended, discuss advantages and disadvantages of existing works and proposed work. Can author compare with works, New group-based generalized interval-valued q-rung orthopair fuzzy soft aggregation operators and their applications in sports decision-making problems.

Aggregation and interaction aggregation soft operators on interval-valued q-rung orthopair fuzzy soft environment and application in automation company evaluation

2. It is better to put values of scores in table of comparisons.

3. Flowchart of method should be added.

4. Language should be improved.

6. PLOS authors have the option to publish the peer review history of their article (what does this mean?). If published, this will include your full peer review and any attached files.

Reviewer #1: No

Reviewer #2: No

---

## [Author Response · Author response to Decision Letter 0]

20 Feb 2024

Response to the Reviewers/Editor in Chief/Associate Editor in Chief of

PLOS ONE

Decision: Revision required [PONE-D-23-39846] - [EMID:e1bac6383c4f50d8] PONE-D-23-39846

Type of manuscript: Article

Manuscript Title: 

“Implementation of Learning Management Systems (LMS) in Higher Education Systems Through Bipolar Complex Hesitant Fuzzy Aczel-Alsina Power Aggregation Operators: A Case Review for China”

Dear Editors and Reviewers:

First, the authors would like to thank the Editor in Chief, Associate Editor, and anonymous referees for spending their time on the manuscript carefully. The comments of the editors and reviewers are valuable. We have taken all the suggestions/comments positively and did our best to incorporate all these suggestions in the revised version. Our pointwise responses to the reviewer’s comments/suggestions are given below.

Reviewer 1 Comments:

Please read my comments/suggestions given below for preparing the revised draft:

My Comments and Suggestions to Authors:

Response to Reviewer 1# 

Dear Sir/Mam, we appreciate your time in handling our paper and providing suggestions for improvement. We believe the quality of the revised version has considerably improved and hope that you find the revised manuscript satisfactory this time.

Comment 1: The abstract is not convincing and is disorganized, it should be refined to precisely illustrate what authors have done in this paper within 200 words. The abstract must be a concise yet comprehensive reflection of what is in your paper. Remember that reader want to know: 1-what is the problem. 2- why the problem is relevant 3- wants an overview of your approach. 4-need to know the results.

Response: Dear Sir/Mam, thank you very much for taking an interest and pointing out these problems. Dear Sir/Mam, we have revised the abstract section by discussing the major problems, why the problem is relevant, an overview of the proposed work, and the major contribution of the proposed theory as per your suggestions and highlighted them in the revised manuscript, I hope this time you will be satisfied.

Comment 2: Manuscript needs a good introduction, the introduction section of the manuscript is weak, authors are advised to improvise the introduction section. In the Introduction part, the new features of the proposed method and the main advantages of the results over others should be clearly described.

Response: Dear Sir/Mam, thank you very much for pointing out these problems. Dear Sir/Mam, we have revised the introduction section by discussing the main advantages of the proposed work as per your suggestion and highlighted in the revised manuscript, I hope this time you will be satisfied.

Comment 3: An introduction should clearly highlight the motivation, problem statement, the objective of the paper, gap in the existing research and the novelty of the conducted research.

Response: Dear Sir/Mam, thank you very much for pointing out these problems. Dear Sir/Mam, we have revised the introduction section by discussing the motivation, problem statement, research gap, novelty, and objectives of the proposed work as per your suggestion and highlighted in the revised manuscript, I hope this time you will be satisfied.

Comment 4: This application topic has not received much attention in the literature. However, the study, literature review and presentation require substantial improvement in several respects.

Response: Dear Sir/Mam, thank you very much for pointing out these problems. Dear Sir/Mam, we have included a paragraph based on the application topic in the introduction section and also improved the literature section, and presentation of the proposed work as per your suggestion and highlighted in the revised manuscript, I hope this time you will be satisfied.

Comment 5: How are the parameters in the proposed model selected?

Response: Dear Sir/Mam, the value of the parameter follows as , but in the proposed application we have used the value of the parameter as , it is dependent on the authors.

Comment 6: The problems in this manuscript are not clear.

Response: Dear Sir/Mam, thank you very much for pointing out these problems. Dear Sir/Mam, we have very briefly discussed the problems in the existing technique in the abstract, introduction, and application section, I hope this time you will be satisfied.

Comment 7: There are no citations for many sentences in this manuscript. Why? Please check.

Response: Dear Sir/Mam, we have done the needful, I hope this time you will be satisfied.

Comment 8: Some variables weren't defined appropriately.

Response: Dear Sir/Mam, thank you very much for pointing out these problems. Dear Sir/Mam, we have briefly defined every variable in the revised manuscript as per your suggestion and highlighted it in the revised manuscript, I hope this time you will be satisfied.

Comment 9: English language needs to be improved significantly.

Response: Dear Sir/Mam, thank you very much for pointing out these problems. Dear Sir/Mam, we have very briefly improved the English language especially grammatical mistakes and typos error as per your suggestion and highlighted in the revised manuscript, I hope this time you will be satisfied.

Comment 10: Paper organization needs to be restructured. Kindly be as concise and straight to the point as possible.

Response: Dear Sir/Mam, thank you very much for pointing out these problems. Dear Sir/Mam, we have restructured the manuscript as per your suggestion and highlighted it in the revised manuscript, I hope this time you will be satisfied. 

Comment 11: The writing of the manuscript. There are a many incomplete sentences or sentences without subjects.

Response: Dear Sir/Mam, thank you very much for pointing out these problems. Dear Sir/Mam, we have done the needful, I hope this time you will be satisfied.

Comment 12: Many details are missing and others unclear.

Response: Dear Sir/Mam, thank you very much for pointing out these problems. Dear Sir/Mam, we have done the needful, I hope this time you will be satisfied.

Comment 13: The database requires more explanation.

Response: Dear Sir/Mam, thank you very much for pointing out these problems. Dear Sir/Mam, we have included more explanations in the revised manuscript as per your suggestions, I hope this time you will be satisfied.

Comment 14: Result and Discussion section is inadequate. Need more attention and better explanation.

Response: Dear Sir/Mam, thank you very much for pointing out these problems. Dear Sir/Mam, we have done the needful, I hope this time you will be satisfied.

Comment 15: When I checked the results, I noticed that there were mistakes, please recheck.

Response: Dear Sir/Mam, thank you very much for pointing out these problems. Dear Sir/Mam, we have done the needful, I hope this time you will be satisfied.

Comment 16: The results are not easy to follow.

Response: Dear Sir/Mam, thank you very much for pointing out these problems. Dear Sir/Mam, we have improved all results in the proposed theory as per your suggestion and highlighted in the revised manuscript, I hope this time you will be satisfied.

Comment 17: I suggest extending the conclusions section to focus on the results you get, the method you propose, and their significance.

Response: Dear Sir/Mam, thank you very much for pointing out these problems. Dear Sir/Mam, we have revised the conclusion section as per your suggestion and highlighted it in the revised manuscript, I hope this time you will be satisfied.

Comment 18: There are many repetitions in sentences.

Response: Dear Sir/Mam, thank you very much for pointing out these problems. Dear Sir/Mam, we have removed the repetition in the sentence as per your suggestion and highlighted it in the revised manuscript, I hope this time you will be satisfied.

Comment 19: References aren't formatted according to rules.

Response: Dear Sir/Mam, thank you very much for pointing out these problems. Dear Sir/Mam, we have done the needful, I hope this time you will be satisfied.

Comment 20: Some articles listed in References are old, we are in 2024.

 Response: Dear Sir/Mam, thank you very much for pointing out these problems. Dear Sir/Mam, we have done the needful, see Ref. [29-37], I hope this time you will be satisfied.

Reviewer 2 Comments:

In this paper author addressed Implementation of Learning Management Systems (LMS) in Higher Education Systems Through Bipolar Complex Hesitant Fuzzy Aczel-Alsina Power Aggregation Operators: A Case Review for China. The paper needs following revision;

Response to Reviewer 2# 

Dear Sir/Mam, we appreciate your time in handling our paper and providing suggestions for improvement. We believe the quality of the revised version has considerably improved and hope that you find the revised manuscript satisfactory this time.

Comment 1: Comparison of the paper should be extended, discuss advantages and disadvantages of existing works and proposed work. Can author compare with works, New group-based generalized interval-valued q-rung orthopair fuzzy soft aggregation operators and their applications in sports decision-making problems. Aggregation and interaction aggregation soft operators on interval-valued q-rung orthopair fuzzy soft environment and application in automation company evaluation.

Response: Dear Sir/Mam, thank you very much for pointing out these problems. Dear Sir/Mam, we have improved the comparison section (see Ref. [29-32]) as per your suggestion and highlighted in the revised manuscript, I hope this time you will be satisfied.

Comment 2: It is better to put values of scores in table of comparisons.

Response: Dear Sir/Mam, thank you very much for pointing out these problems. Dear Sir/Mam, we have done the needful, I hope this time you will be satisfied.

Comment 3: Flowchart of method should be added.

Response: Dear Sir/Mam, the needful is done.

Comment 4: Language should be improved.

Response: Dear Sir/Mam, thank you very much for pointing out these problems. Dear Sir/Mam, we have very briefly improved the English language especially grammatical mistakes and typos error as per your suggestion and highlighted in the revised manuscript, I hope this time you will be satisfied.

---

## [Decision Letter · Decision Letter 1]

27 Feb 2024

Implementation of Learning Management Systems (LMS) in Higher Education Systems Through Bipolar Complex Hesitant Fuzzy Aczel-Alsina Power Aggregation Operators: A Case Review for China

PONE-D-23-39846R1

Dear Dr. Yin,

We’re pleased to inform you that your manuscript has been judged scientifically suitable for publication and will be formally accepted for publication once it meets all outstanding technical requirements.

Kind regards,

Van Thanh Tien Nguyen, Ph.D.

Academic Editor

PLOS ONE

Additional Editor Comments (optional):

Reviewers' comments:

Reviewer's Responses to Questions

**Comments to the Author**

1. If the authors have adequately addressed your comments raised in a previous round of review and you feel that this manuscript is now acceptable for publication, you may indicate that here to bypass the “Comments to the Author” section, enter your conflict of interest statement in the “Confidential to Editor” section, and submit your "Accept" recommendation.

Reviewer #1: All comments have been addressed

Reviewer #2: All comments have been addressed

2. Is the manuscript technically sound, and do the data support the conclusions?

Reviewer #1: Yes

Reviewer #2: Yes

3. Has the statistical analysis been performed appropriately and rigorously? 

Reviewer #1: Yes

Reviewer #2: Yes

4. Have the authors made all data underlying the findings in their manuscript fully available?

Reviewer #1: Yes

Reviewer #2: No

5. Is the manuscript presented in an intelligible fashion and written in standard English?

Reviewer #1: Yes

Reviewer #2: Yes

6. Review Comments to the Author

Reviewer #1: The major contribution of the proposed theory is listed below:

1) We derived the techniques of BCHF sets, then we evaluated some flexible operational laws,

called Algebraic operational laws and Aczel-Alsina operational laws.

2) We elaborated on the techniques of BCHFAAPA, BCHFAAPWA, BCHFAAPG, and

BCHFAAPWG operators.

3) We discussed some basic properties of each proposed operator.

We computed the MADM techniques for invented operators.

4) We selected some prevailing operators and tried to compare their ranking results with our

proposed results to enhance the worth and capability of the invented theory

I accept this paper.

Reviewer #2: The paper can be accepted now. Author has revised paper carefully and addressed each comments of the reviewers.

7. PLOS authors have the option to publish the peer review history of their article (what does this mean?). If published, this will include your full peer review and any attached files.

Reviewer #1: No

Reviewer #2: No

---

## [Editor Report · Acceptance letter]

21 Mar 2024

PONE-D-23-39846R1 

PLOS ONE

Dear Dr. Yin, 

I'm pleased to inform you that your manuscript has been deemed suitable for publication in PLOS ONE. Congratulations! Your manuscript is now being handed over to our production team.

Kind regards, 

on behalf of

Ass. Prof. Van Thanh Tien Nguyen 

Academic Editor

PLOS ONE